# Molecularly Imprinted Nanomaterials with Stimuli Responsiveness for Applications in Biomedicine

**DOI:** 10.3390/molecules28030918

**Published:** 2023-01-17

**Authors:** Yan Zhang, Qinghe Wang, Xiao Zhao, Yue Ma, Hongbo Zhang, Guoqing Pan

**Affiliations:** 1School of Chemistry and Chemical Engineering, Jiangsu University, 301 Xuefu Road, Zhenjiang 212013, China; 2Institute for Advanced Materials, School of Materials Science and Engineering, Jiangsu University, 301 Xuefu Road, Zhenjiang 212013, China; 3College of Life Sciences, Northwest Normal University, Lanzhou 730071, China; 4Pharmaceutical Sciences Laboratory, Åbo Akademi University, 20520 Turku, Finland; 5Turku Bioscience Centre, University of Turku and Åbo Akademi University, 20520 Turku, Finland

**Keywords:** stimuli responsiveness, molecularly imprinted polymers, nanomaterials, biomedical science

## Abstract

The review aims to summarize recent reports of stimuli-responsive nanomaterials based on molecularly imprinted polymers (MIPs) and discuss their applications in biomedicine. In the past few decades, MIPs have been proven to show widespread applications as new molecular recognition materials. The development of stimuli-responsive nanomaterials has successfully endowed MIPs with not only affinity properties comparable to those of natural antibodies but also the ability to respond to external stimuli (stimuli-responsive MIPs). In this review, we will discuss the synthesis of MIPs, the classification of stimuli-responsive MIP nanomaterials (MIP-NMs), their dynamic mechanisms, and their applications in biomedicine, including bioanalysis and diagnosis, biological imaging, drug delivery, disease intervention, and others. This review mainly focuses on studies of smart MIP-NMs with biomedical perspectives after 2015. We believe that this review will be helpful for the further exploration of stimuli-responsive MIP-NMs and contribute to expanding their practical applications especially in biomedicine in the near future.

## 1. Introduction

### 1.1. Molecularly Imprinted Polymers

Molecular imprinting technology (MIT) is a method to synthesize plastic antibodies that simulate the specific recognition characteristics of targeted biomolecules, such as receptors and antibodies. The prepared plastic antibodies, which are called molecularly imprinted polymers (MIPs), can provide selective sites for the specific recognition of template molecules. MIT first appeared in immunology in the 1940s, when Nobel Prize winner Linus Pauling proposed the theory of antibody formation [1]. In 1949, Frank H. Dickey developed the concept of “specific adsorption”, which greatly promoted the generation of molecular imprinting [2]. However, real MIPs were not successfully prepared until 1972 by Günter Wulff’s research group [3]. The first paper using the vocabulary of the imprinted polymer was written in 1985 [4], which led to a breakthrough in MIT research. At that time, MIT developed rapidly and became a new type of chemical analysis technology. In 1993, Mosbach et al. reported a molecular imprinting study of theophylline [5], confirming that imprinted polymers exhibited similar characteristics to antibodies. In just a few decades, MIT has developed rapidly and aroused great interest. So far, the application of MIT has covered a wide range of fields, including biomimetic sensing, simulated enzyme catalysis, chromatographic separation, solid-phase extraction, clinical drug analysis, etc. [6,7,8,9,10,11].

MIPs prepared by MIT have imprinted cavities with complementary functional groups, shapes, and sizes towards template molecules, which can be utilized to specifically recognize target molecules [12]. The general method for preparing MIPs is firstly to realize the pre-assembly of template molecules and functional monomers in a solution through covalent or non-covalent (hydrogen bond, coordination bond, electrostatic interaction, etc.) combinations, and then to form cross-linked polymers via polymerization to finally obtain MIPs after the template is removed (Figure 1) [13]. Because of its high stability, simple preparation, low cost, and other advantages, MIT has become a research hotspot [14]. In addition, MIPs are particularly well studied in the biomedical fields due to their similar functional characteristics with biological antibodies and more flexible designability. Although the advanced research of molecular imprinting has not been practically applied to clinical medicine, with skillful design, MIPs still have good development prospects and application potential in the near future.

### 1.2. Molecularly Imprinted Polymer Nanomaterials (MIP-NMs)

With the deepening of MIT research, MIPs with different shapes and sizes have been developed so far. In the early days, traditional bulk polymerization, precipitation polymerization, emulsion polymerization, and other methods were usually used to prepare bulk polymers (the size of these particles is generally in the range of 10^−2^–10^−6^ m) or micron-sized particles [15,16]. Bulk particles prepared by traditional bulk polymerization cannot be used directly because of their uneven shape and large size, and need to be ground and sieved to achieve the desired size. MIPs prepared by this method, due to the destruction of part of the imprinted sites, result in a decreased specific binding capacity [17]. Micron-sized MIPs synthesized by traditional precipitation polymerization and emulsion polymerization also have shortcomings, such as difficulty in the elution of template molecules, the limited number of MIP surface binding sites, a poor ability to specifically recognize template molecules, poor dispersion, etc. However, nanostructured materials, a rapidly developing field, provide practical methods for solving these problems.

In recent decades, breakthroughs have been made in the preparation of advanced functional nanomaterials, and researchers have devoted themselves to the study of molecularly imprinted polymer nanomaterials (MIP-NMs) (diameters less than 200 nm) with a uniform particle size and controllable morphology. Compared with traditional MIPs, MIP nanoparticles (MIP-NPs) have a good dispersion ability, a high surface-area-to-volume ratio, and easy elution of template molecules, as well as exhibit a high binding capacity, high selectivity, high affinity, and good water compatibility [18]. Therefore, the application fields of MIP-NPs have also expanded from separation and detection to biomedicine (bioimaging [19,20,21,22], cancer diagnosis [23,24,25], targeted drug delivery [26,27], controlled drug release [28,29,30], etc.). MIP-NPs are expected to become an attractive research tool for a wide range of studies in the biomedical field. Molecularly imprinted nanogels (MIP-NGs) are also a promising innovative material in the biomedical field. As a dispersion based on swollen hydrogel nanoparticle networks composed of hydrophilic or amphiphilic polymer chains, MIP-NGs have relatively high uniformity, adjustable sizes, low toxicity, stability in serums, and good drug encapsulation ability, etc.; hence they are referred to as next-generation drug delivery systems [31]. Therefore, MIP-NGs show great potential in disease diagnosis, cancer drug delivery, and imaging.

Since MIP-NMs have comparable selectivity and affinity to natural receptors, their applicability becomes wider and wider in biomedicine. In recent years, the template molecules used in the preparation of nanomaterials are not only active small molecules but also biological macromolecules, such as macromolecular proteins, cells, and various microorganisms, viruses, etc., which are introduced in detail in the following review.

#### 1.2.1. Small-Molecule-Imprinted Nanomaterials

Nanomaterials imprinted with small organic molecules (drug molecules, amino acids and their derivatives, carbohydrates, and flavonoids) are the most common molecularly imprinted nanomaterials, mainly due to their similar prepared strategies to traditional molecularly imprinted nanomaterials. During the preparation process, typical active small-molecule template molecules can successfully form a complex with functional monomers because of their physical and chemical stability. Moreover, the ratio of the template and monomer can be accurately regulated in the imprinting process.

##### Drug Molecule Imprinting

Nowadays, MIP-NMs have been widely used in various fields of drug research, such as the separation and purification of target substances, the extraction of active ingredients from a drug [32], and their application to drug delivery systems to achieve controlled or sustained drug release. Therefore, many drug molecules are introduced as template molecules, such as paracetamol, ofloxacin, the antidepressant sertraline, and carbamazepine [33,34,35,36].

What is worth discussing here is the chiral drug molecule, which is usually used as a template. The so-called chiral drug molecule has a chiral center in the drug’s molecular structure [37]. Although the physicochemical properties of this pair of isomers are not much different, their biological activities are quite diverse [38]. They often have different pharmacological effects, metabolic processes, and toxicities—for example, out of the typical chiral drug molecules tiracetam and levetiracetam, only levetiracetam has an antiepileptic effect [39]. Another example is carnitine and L-carnitine; L-carnitine can improve cell metabolism, while carnitine can cause damage to muscles and has adverse side effects [40]. However, it is hard for the traditional method to separate chiral molecules, and it is of great significance to use MIP-NMs to separate single isomers in chiral molecules. For example, Goyal et al. prepared a chiral drug molecule (S)-naproxen surface-imprinted layer on magnetic nanoparticles via precipitation llymerization, which can be used to separate the corresponding enantiomers from the racemic mixture (R, S)-naproxen (Figure 2) [41]. The results showed that it has more affinity towards naproxen than other chemically similar drugs, and enables the selective isolation of (S)-naproxen from the corresponding racemic mixture, with the binding capacity 4.1 times more than that of (R)-naproxen. Such multi-reusable MIP-NPs could potentially be useful for the separation of new chiral molecules at a much lower cost in a relatively shorter time.

##### Small-Biomolecule Imprinting

The main biomolecules used as template molecules to synthesize MIP-NMs include short peptides, monosaccharides, etc. In recent years, scientific research has found that the main form of protein absorption by the human body is not through amino acids, but in the form of short peptide absorption. The recognition of short peptides is a subject that governs many fundamental biological processes, and MIT is a simple and effective method for short peptide recognition. Zeng et al. prepared MIP-NPs with an affinity for the hydrophilic peptide GFP-9 by inverse microemulsion polymerization [42]. The surface has a high-affinity peptide binding site and strong ability to bind to the template molecule GFP-9. For short peptide imprinting, the most important thing is to use it as an epitope to realize the corresponding macromolecular recognition. Lucia et al. selected the heart failure biomarker protein Troponin I, and used its specific sequence NR11 as a template [43]. Free-radical-initiated precipitation polymerization was carried out to prepare a nano-MIP with a size of about 60 nm, which has a single binding site on every MIP nanoparticle and shows a recognition performance comparable to that obtained via solid-phase synthesis. They also investigated the effect of the total monomers-to-template molar ratio on the formation of the binding sites, as well as the pivotal role of hydrophobic interactions on the formation of nanoscale particles. This work determined the conditions for preparing synthetic recognition nanoparticles acting as peptide-recognition elements. Li et al. fabricated N-isopropyl acrylamide (NIPAAm)-based epitope surface imprinting sites on SiO_2_ nanoparticles by using the His-tag-anchored epitope of human serum albumin (HSA) as templates, which could capture the HSA at 45 °C and release it 4 °C lower with an imprinting factor of 4.0 [44]. Teixeira et al. found that growth factor-β3 (TGF-β3) MIP-NPs, synthesized via epitope-imprinting inverse microemulsion polymerization, can recognize and preferentially bind TGF-β3 even from a complex human fluid (platelet lysate). Additionally, culturing adipose-derived stem cells with MIP-NPs incubated with platelet lysate led to a higher collagen II-rich matrix, which can be used as cost-effective, stable, and scalable alternative abiotic GF ligands to guide cell fate [45]. Moreover, it was verified that MIP-NPs can be used as GF chelating ligands to regulate cell behavior, and the functional surface of MIP-NPs can specifically recognize and capture TGF-β3 and 3D cell cultures using MIP-NPs, which can promote the deposition of chondro-related matrix components.

MIPs also can be utilized to measure the glucose content in blood or urine, which is a simple and efficient research method for detecting blood sugar. So, most glucose-imprinted polymers are used to construct sensors with more and more accurate measurements of glucose in biosamples. Recently, Caldara et al. used them as a recognition element for the thermal detection of glucose in urine [46]. They introduced a virtual imprinting method, selecting glucuronic acid (GA), acrylamide (AAM), ethylene glycol dimethacrylate (EGDMA), and AIBN as a pseudo-template molecule, functional monomer, cross-linking agent, and thermal initiator, respectively, to synthesize MIP-NPs, which were integrated into a thermally conductive acceptor layer via microcontact deposition to analyze the glucose levels in human urine samples. Some other monosaccharides are also biomarkers of special cells; therefore, the MIP-NMs with those monosaccharide templates can be applied in the field of cell recognition. For example, Yin et al. synthesized a sialic-acid-imprinted polymer layer coating on mesoporous SiO_2_ nanoparticles for cancer cell recognition and bio-imaging [47]. They verified that the combination of mesoporous nanocarriers with SA recognition sites gives MIP-NPs the ability to target SA-overexpressed cancer cells and enhance cancer cellular uptake, finally leading to the improved inhibition of cancer cell proliferation.

Moreover, some biomolecules, such as amino acid residue and phosphorylated amino acid residue, were also used as templates in molecular imprinting to monitor life activities. For example, phosphorylated tyrosine residue-imprinted mesoporous materials were reported by Zhang et al. for the first time. They used the epitope of tyrosine phosphorylation, phenylphosphonic acid (PPA), as a dummy template to prepare the molecularly imprinted mesoporous material (MIM), with an excellent affinity towards pTyr. More importantly, the dummy template, PPA, could effectively prevent the false positive results brought about by template leakage [48]. There are many preparation methods of MIP-NMs for small biological molecules that are imprinted, such as in situ polymerization, surface imprinting, precipitation polymerization, suspension polymerization, etc. [49]. Each method has unique advantages and disadvantages, such as the simple preparation and convenient operation of precipitation polymerization, and the obtained MIPs are uniform in shape and easy to manipulate in size, but the template molecules inside them are not easily eluted. Moreover, when the target molecule is specifically recognized in the solution, the residual template molecule in the MIPs may fall off, resulting in experimental errors. Therefore, when preparing MIP-NMs, the specific method used depends on the actual situation.

#### 1.2.2. Biomacromolecule-Imprinted Nanomaterials

With the development of MIT, small-molecule MIPs are relatively mature, and the recognition mechanism is relatively clear, but the research on macromolecules and complex molecules is still in the development stage. Small molecules diffuse easily and can easily approach the imprinting site, while, in contrast, the functional structure of biological macromolecules is more complex. Their effective imprinting sites may be located on the surface of the MIPs or in the pore; thus the protein embedded inside cannot be eluted and the inner imprinting sites do not work. Moreover, the configuration of biomacromolecules may change during polymerization, which makes MIPs mismatched or have a low affinity for the template macromolecule [50]. Therefore, the imprinting method for biological macromolecules is also different from that of small molecules. With the continuous development of molecular imprinting technology, recent research has gradually turned to surface imprinting and epitope imprinting. In addition, solid-phase imprinting is also gradually becoming popular; the following is a detailed overview.

##### Polysaccharide Imprinting

In the human body, some polysaccharides have anti-tumor effects, and some polysaccharides have good hypoglycemic effects. In addition, some polysaccharides have anti-aging, anti-oxidation, and anti-ulcer activities and regulate body immunity. For example, heparin in the human body is composed of two multimers made of alternately linked polysaccharides with an anticoagulant effect [51]; hyaluronic acid can regulate vascular permeability and promote wound healing [52], etc. Therefore, it can be seen that polysaccharides play an extremely important role in the human body, and their separation and detection are of great significance. For example, Li et al. prepared new magnetic MIPs with two template molecules (fucoidan and alginic acid) for the recognition of algal polysaccharides [53]. In addition, polysaccharides also exist on the surface of the cell membrane, so the preparation of nanomaterials with carbohydrate molecules as template molecules also opens up a new way for cancer treatment [54]. For glycan imprinting, the main obstacle is the macromolecular structure. Several efficient techniques for imprinting macromolecular polysaccharides have been developed, such as surface imprinting, molecularly imprinted cryogel, and epitope imprinting [50].

##### Nucleic Acid Imprinting

Nucleic acid is a biological macromolecular compound composed of a variety of nucleotides, and it is also one of the most important living substances in the human body. Deoxyribonucleic acid (DNA) is crucial for determining the genetic information of species. Unlike other polymers, DNA sequences can be precisely controlled. DNA is not only a kind of genetic material; functional DNA can realize molecular recognition and catalysis [55]. MIPs with structure specificity and predictability also make great contributions in the field of nucleic acid research, especially in the significance of DNA sensors for the detection of genetic diseases.

In the 1880s, Piletskiy et al. conducted preliminary studies on imprinted nucleic acids [56]. This early study showed that molecules such as nucleotide triphosphates, nucleotides and nucleosides may help to make polymeric structures of complementary templates with different specificities, which opened the door to imprint DNA in the field of molecular imprinting. So far, a variety of blotting methods have been applied to identify and detect nucleic acid targets, including small molecular bases, single-stranded DNA (ssDNA), double-stranded DNA (dsDNA), and long-stranded DNA [57]. For example, Slinchenlo et al. used epitope imprinting to prepare double-stranded DNA (dsDNA)-imprinted polymers on the surface of silanized glass slides for the first time, using a bacterial exotoxin gene double-stranded DNA as a template molecule, 2-vinyl-4,6-diamino-1,3,5-triazine (VDAT) and acrylamide as functional monomers, and methylenebisacrylamide (BIS) as a cross-linking agent [58]. The results showed that the prepared imprinted polymers coated on glass slides can indeed be used as a selective surface for the detection of the desired DNA sequences. Brahmbhatt et al. successfully obtained MIP-NPs with an oligomer DNA sequence (5′-AGC TAG CTA GCT-3′) as a template molecule using a solid-phase imprinting polymerization strategy (Figure 3) [59]. Introducing modified thymine bases into the oligonucleotide sequence allowed for the establishment of covalent anchoring points between the oligonucleotide and polymer matrix, which is better than that anchored by Acrydite^TM^. It was also demonstrated that the nucleotide hybrid MIP-NPs could specifically recognize their target DNA. Moreover, the introduction of complementary DNA strands as “pre-organized selective monomers” improved the recognition performance without affecting the physical properties of the nanoparticles, such as size, shape, or dispersion.

Compared with DNA, the structure of RNA is more complex, so the difficulty of imprinting RNA is also increased. In recent years, the research on DNA has been in depth, while the analysis and detection of RNA need to be improved.

##### Protein Imprinting

Protein is the most common biomolecule in the body, which is crucial to living processes. The separation, detection, and analysis of proteins are of great significance in the biomedical field. Therefore, protein-imprinted polymer nanomaterials have been widely studied. In 1996, Hjerten et al. synthesized low cross-linking gels with acrylamide as the functional monomer for the first time, and imprinted hemoglobin, growth hormone, myoglobin, and ribonucrenase by the embedding method, obtaining MIPs with specific selection, which opened the study of imprinted protein macromolecules [60]. Since then, more and more researchers have devoted themselves to the study of protein macromolecular imprinting technology, and the imprinting method has also expanded from the simple embedding method.

##### Surface Imprinting

The earliest use of surface imprinting to prepare imprinted protein MIPs was reported by Glad et al., who reacted the glycoprotein transferrin with boronate silane in a solution to polymerize on the surface of porous silica [61]. The boronate group can reversibly react with sugars and glycoproteins through the ester group, and the pre-binding of boronate silane with transferrin makes the boronate group correctly arranged, so the imprinted site has specific binding to transferrin. In 1999, Shi et al. proposed the plate surface imprinting method, which firstly adsorbed proteins on mica with high atomic contents, then coated disaccharide molecules on proteins, and the two were bonded by a hydrogen bond. Then a thin layer of fluorescent polymer was polymerized on the surface of sugar molecules. Finally, the mica were removed and the protein molecules were dissolved. A polydisaccharide surface-imprinted polymer with protein-shaped holes was generated [62]. In recent years, efforts have been devoted to the preparation of MIPs with core/shell structures, which can be divided into two kinds by their different sequences of protein addition: prefixed and unfixed (Figure 4). Li et al. used boronate affinity to fix glycoprotein with controllable orientation first, and then the self-polymerization of 2-anilinoethanol was carried out to form a thin imprinting layer on the surface of the magnetic nanoparticles [63]. Moreover, the interaction between aptamer and BSA was also chosen to immobilize the BSA on the surface of nanoparticles for temple-prefixed surface imprinting (Figure 4a) [64]. Fu’s group developed a series of surface imprinting SiO_2_ nanoparticles with protein dissolved in the reaction solution [65]. For example, they developed a novel strategy for pegging surface-imprinted nanoparticles using surface-induced reversible addition–fragmentation chain transfer with lysozyme as a template (Figure 4b) [66]. Water precipitation polymerization (SI-RAFT APP) was used to synthesize the imprinted nanoparticles with a core/shell structure, then nonlinear PEG chains were grafted onto their surfaces, and finally the template was removed. The thickness of the imprinted nano-shells and the length of the grafted chains could be controlled by the polymerization time.

##### Epitope Peptide Imprinting

Epitope imprinting technology has also made remarkable progress. Due to the large molecular weight of protein macromolecules, steric resistance, and thermodynamic effects, methods such as surface imprinting may cause the template molecules to be eluted uncleanly, so some protein residues may affect the experimental results. Moreover, large imprinting sites may have more interactions with a series of other small molecule peptides, which may hinder or weaken the imprinting effect and reduce its selectivity, resulting in the poor recognition of target proteins by synthetic MIPs [67]. Therefore, the imprinting effect of small biomolecules tends to be better. Epitope imprinting makes up for this shortcoming.

Epitope imprinting uses a short peptide consisting of 9 to 15 amino acid residues exposed in the surface structure of the protein as a template molecule. Only this short peptide is imprinted, and the obtained MIPs cannot only recognize the short peptide but also specifically recognize the entire protein with the same peptide chain [68]. This method is inspired by nature; that is, when an antibody recognizes an antigen, it only binds to a part of the epitope. This method was proposed by Rachkov et al. as early as 2001 [69]. To explore the epitope approach, they chose to identify the macromolecule neurohypophyseal hormone, oxytocin, by using its surface tetrapeptide, a short peptide with the same structural fragments as the macromolecule, as a template to produce MIPs with specific recognition of oxytocin. The results show that the epitope imprinting method is feasible, short-peptide molecules are relatively cheap, and MIPs are easier to prepare with higher selectivity and adsorption. Recently, epitope imprinting has prospered and been carried out via different preparation methods, such as precipitation polymerization [44], inverse emulsion polymerization [45], solid-phase synthesis [70], and so on (Figure 5). For example, the peptide (YVGSKTKEGVVHGVA) from α-synuclein (SNCA) was used as the epitope to construct magnetic molecularly imprinted composite nanoparticles (MMIPs) in Lee’s study [71]. Moreover, these MMIPs were applied to treat brain endothelial cells and remove α-Synuclein from an in vitro model (Figure 5a). Teixeira et al. designed the conformational epitopes of transforming growth factor-β3 (TGF-β3) and imprinted the surface onto polyacrylamide-based nanoparticles by reverse microemulsion polymerization (Figure 5b) [45]. Mier et al. reported the research of MIP-NGs for the selective recognition of HAVCR-1 protein. They used a computer-based rational approach to make sure of the epitope template [70]. Then the epitope peptide was oriented and immobilized on glass beads by strain-promoted alkyne-azide cycloaddition for the first time in solid-phase imprinting. They also confirmed and gained further insights into the affinity of MIP-NGs toward templates by solution STD and WaterLOGSY NMR spectroscopies (Figure 5c). In other words, the epitope method changes the complicated protein imprinting into the relatively simple small-molecule imprinting, with a performance closer to the recognition of natural antibodies [42,43].

Except for epitope peptide imprinting, the part of protein can be also chosen as the template in some protein systems with special structures. Such a fragment imprinting strategy in precipitation polymerization facilitates the orientation of the recognition proteins and keeps the active domain outside. Natsuki et al. prepared molecularly imprinted polymer nanogels (MIP-NGs) imprinted with the fragment crystallization (Fc) domain of immunoglobulin G (IgG) [72]. The unique domain recognition property resulted in the suppression of the immune response in Fc domain receptor-possessing macrophages and natural killer cells due to the regulation of protein coronas based on the oriented adsorption of IgG.

##### Solid-Phase Imprinting

Currently, the solid-phase imprinted synthesis of MIP-NMs has been shown to be a state-of-the-art method for synthesizing receptors for a range of diagnostic and biomedical applications, usually by first immobilizing template molecules on supports (glass beads) before adding reactive monomers to induce polymerization [73]. After adding an initiator to initiate polymerization, the unreacted monomers or low-affinity polymers are washed away. The obtained MIP-NMs are attached to the solid support, and then the MIP-NMs are washed off using a suitable condition [74,75]. It can be carried out by persulfate-initiated polymerization in an aqueous solution or UV-initiated in an organic solvent. To be effectively used in the medical field, the polymerization should be carried out in an aqueous solution as much as possible. Studies have shown that the process is easy to automate and can yield well-performing MIP-NMs in a relatively short period. Due to its affinity purification step, the solid-phase synthesis of MIP-NMs has a high affinity and strong specificity, and the binding sites appear uniformly distributed. Since the template molecule is immobilized on the solid support and can be reused, the template molecule is no longer required when preparing MIP-NMs to pass through the polymerized monomer, which reduces the number of template molecules used, and at the same time, the tedious template elution step is no longer required. Based on the solid-phase imprinting method, Xu et al. used functionalized glass beads (GB) with metal chelates for the immobilization of trypsin [76]. The thermos-responsive functional monomer N-isopropyl acrylamide (NIPAAm), the anchor monomer p-aminobenzamidine (PAB) of the affinity ligand, and the cross-linker EbAm were introduced at 37 °C under the conditions of the initiator system. This solution polymerized and eluted MIP-NPs at 25 °C. The results showed that the MIP-NPs synthesized by solid-phase blotting could be stored stably for up to 6 months in a buffer at 4 °C and had a higher binding affinity and higher selectivity for the target molecule, trypsin (Figure 6a).

Moreover, Sellergen’s group reported a new approach similar to solid-phase imprinting for the synthesis of protein-imprinted nanogels [77]. Compared with normal solid-phase imprinting, they anchored trypsin on the nanosized magnetic template carriers, and the imprinting process was produced by the high-dilution polymerization of monomers in the presence of the magnetic template. This protocol produced MIP-NGs in high yields and with a template-free form (Figure 6b).

#### 1.2.3. Cell Imprinting

The imprinting of cells requires milder conditions and a pretreatment of the cells, so cellular imprinting is the most challenging form of molecular imprinting, and whole-cell imprinting is a milestone in biological macromolecular imprinting. In 1996, Vulfson et al. used listeria and staphylococcus aureus as templates to imprint whole cells on the surface of polyamide microspheres for the first time [78]. In 2002, Dickert et al. imprinted yeast cells [79]. The most commonly used research method for imprinting whole cells is microcontact printing, in which template cells are fixed on the substrate support layer, then covered, cured with a monomer or soft polymer, sandwiched between the support layer and curing polymer matrix, and finally moved to create a template-fixed support layer of the cells, obtaining MIPs with imprinted loci. It is worth noting that if you want to apply cell-imprinted MIPs to cell recognition in vivo, you need to prepare MIPs with a size similar to that of natural macromolecules, that is, MIP-NMs, to facilitate diffusion and circulation in blood vessels, lymphatic systems, and other spaces. However, imprinting whole cells often yields micro-MIPs, which are not suitable for in vivo studies [80]. In addition, direct imprinting is very complex due to the large size and the soft membrane of mammalian cells, and such cells are diverse in their composition and complex in their structure; it is difficult to differentiate between different cell lines solely by shape imprinting. Most of the MIP-NMs are used for imprinting and recognizing part of the special cell membrane.

Therefore, a method to replace the imprinting of the whole cell was proposed, which entails selecting the surface components of the cell membrane (saccharide derivatives, proteins, lipids, etc.) as template molecules for imprinting, which not only avoids the use of the cell in the imprinting processes but also realizes final cell recognition. One approach is to use sugar structures of the cell membrane surface as template molecules. Recently, Lu et al. targeted sialic-acid-overexpressing tumor cells by imprinting sialic acid [81]; Piletsky et al. prepared MIP-NPs by solid-phase imprinting with B-type blood trisaccharide as a template, which can efficiently and specifically differentiate red blood cells of different blood types [82]. Another approach is to use cell membrane surface proteins or corresponding polypeptides as targeted template molecules for imprinting cells. Cancer cells overexpress the epidermal growth factor (EGFR), so Wang et al. used epitope imprinting to prepare dual-template MIP-NPs with EGFR and doxorubicin (DOX) for targeted imaging and targeted therapy [83]. The resulting imprinted cavities can be used to identify the corresponding proteins, thereby enabling MIP-NPs to actively target HER2-positive breast cancer cells, effectively targeting and recognizing cancer cells and releasing drugs to kill the breast cancer cells. Therefore, it can be concluded that microcontact and epitope imprinting are generally the two methods that dominate cellular imprinting. Microcontact imprinting is well suited for producing cell-specific recognition surfaces, but epitope imprinting can be used to produce MIP-NMs that can be targeted to target cells in vivo.

#### 1.2.4. Others

In addition to the biomolecules in the human body described above, microorganisms and viruses are also involved in MIT. As synthetic receptors, MIPs have the advantages of good stability, low costs, and environmental friendliness, and can be used for imprinting microorganisms and viruses to achieve microbial detection.

Microbial and viral imprints are also more complex, and their specific surface structures have specific requirements for recognition elements. For the capture of microorganisms, molecular imprinting can be applied to construct polymers with specific binding sites for specific cell membrane/wall components or cell membrane/wall groups [84]. Microbial recognition by MIPs can be divided into two parts, including whole-cell-based molecular imprinting strategies and cell wall component (proteins, lipids, glycans) imprinting strategies. Among the many imprinting methods, the direct imprinting method is the most convenient and simple, which is widely used in microbial imprinting. It is worth noting that direct imprinting is not suitable for some fragile templates, large templates, or unstable molecules. For these microorganisms, indirect imprinting can also be used. Epitope imprinting is also an effective imprinting method. The future development of microbial analysis cannot be underestimated, especially with the continuous development of MIT in recent years. Imprinted microorganisms are not only applied in microbial analysis but are also of great significance when tracking, detecting, and treating diseases caused by microorganisms, such as viruses and bacteria, which greatly enriches the research connotation of MIT.

Most of the MIPs prepared by using the traditional methods described above are rigid structures and can only identify target molecules simply and mechanically. Although molecular imprinting technology has developed rapidly in recent years, compared with the interactions of natural antigens–antibodies and ligands–receptors in vivo, it still lacks flexibility and variability, and cannot actively respond to external stimuli and make corresponding responses. To this end, intelligent stimuli-responsive MIPs came into being.

## 2. Stimuli-Responsive MIP-NMs

Stimuli-responsive polymers have attracted extensive attention due to their controllability, reversibility, and ease of manipulation. In recent years, with the continuous development of smart responsive polymers and MIPs, people have combined these two technologies to synthesize smart stimuli-responsive MIPs (SR-MIPs) [85]. This new group combines the best of both worlds, and the resulting smart SR-MIPs act like switches that can change their physicochemical properties by sensing external stimuli (temperature, light, electricity, magnetism, pH, etc.), to achieve intelligent responsiveness, and the ability to reversibly uptake and release molecules can be obtained under alternating stimuli [86]. Due to their good responsiveness, MIPs have achieved intelligent, switchable recognition characteristics and a controllable high-binding performance.

SR-MIPs are currently at the forefront of MIT. The synthesis processes of these SR-MIPs can generally be divided into two types. One approach is to modify the raw materials or functional monomers to introduce intelligent groups that respond to stimuli; the other approach is to composite other materials with response. The former will affect the networks of MIPs or the recognition of imprinted sites, which will be discussed in detail. The latter is relatively simple, which will be introduced in other parts.

Research has focused on developing stimuli-responsive biological MIP-NMs to mimic human immune responses, such as endogenous stimuli (e.g., reactive oxygen species, changes in physiological pH and temperature, overexpressed protein and enzyme levels) or exogenous stimuli (e.g., temperature, light, changes in the magnetic field, etc.) (Figure 7), and responding to their recognition ability through changes in the environment to simulate the dynamic antigen–antibody or bioreceptor–ligand interaction effects in an organism. In the following, we mainly introduce research in the field of biomedicine having to do with the thermal response, pH response, light response, and so on, which are the most commonly used.

### 2.1. Thermal Responsiveness

Thermal responses are common in SR-MIPs, and thermally responsive MIPs are generally prepared by introducing temperature-sensitive functional monomers, such as NIPAAm [87,88], AAm [89], N, N-dimethyl acrylamide (EDA) [90], etc., among which the construction of imprinted polymers based on the introduction of NIPAAm thermosensitive polymers is a typical imprinting system. As early as 1986, Pelton prepared a monodisperse latex dispersion synthesized by NIPAAm, AAm, and BIS. NIPAAm was demonstrated for the first time to be temperature sensitive [91]. Since then, NIPAAm has been widely used as a representative thermosensitive monomer.

NIPAAm contains a hydrophilic amide group and a hydrophobic isopropyl group, which enables the molecule to exhibit hydrophilic–hydrophobic interconversion performance under relatively mild conditions to changes in external temperature. The low critical solution temperature (LCST) of the polymer prepared by using NIPAAm as a thermosensitive monomer in an aqueous solution is about 32 °C. When the temperature is lower than this critical temperature, the amide group forms hydrogen bonds with water and exhibits hydrophilic ability. Above the critical temperature, the volume of the polymer shrinks so that the water is excluded from the interior. Watanabe et al. combined the unique phase transition mechanism of NIPAAm with MIT for the first time to obtain temperature-sensitive and controllable MIPs, showing that the adsorption amount of template molecules is affected by temperature [92]. After that, NIPAAm could be used as a thermo-responsive monomer in the MIP framework or a switch for recognition on the MIP’s surface. For example, PNIPAAm brushes on the MIP’s surface shrank and blocked the binding sites when the temperature was higher than their LCST [93]. Such stimuli responses did not change the volume of the binding sites, but another strategy, using PNIPAAm in the MIP framework, could change the size of the binding sites to control the affinity of MIPs. Zhang et al. synthesized magnetic thermally responsive MIP-NPs using NIPAAm and AAm, 2-aminoethyl methacrylate (AEM) as functional monomers [94]. This method utilized the thermal responsiveness of surface-imprinted polymers to adjust the binding ability of the MIP-NPs to the template molecule heparin oligosaccharide by regulating the temperature.

In addition, due to the properties of PNIPAAm, it not only tends to control the affinity but also to remove templates or enhance drug holding by its hydrophobicity. For example, by adding NIPAAm to monomers used in the solid-phase imprinting method, MIPs with different affinities at different temperatures can be eluted by controlling the temperature, resulting in high-affinity MIPs, which is due to the volume change in PNIPAAm at different temperatures. Pan’s group used NIPAAm and AAm monomers to cross-link the sialic acid (SA)-imprinted hydrogel layer on mesoporous silica nanoparticles (MSN) by an MBAAm cross-linking agent for the first time to prepare a drug nanocarrier capable of recognizing SA; the typical therapeutic drug doxorubicin (DOX) was filled into mesoporous channels of mesoporous nanocarriers (SIMN) to obtain DOX-loaded nanomaterials (SIMNs@DOX) [47]. The molecularly imprinted nanomaterials showed selective and targeted drug delivery for cancer. The imprinting layer synthesized by introducing the thermo-responsive monomer NIPAAm not only provided a useful encapsulation effect for the drug but also reduced the risk of drug leakage during the detection process. Compared with natural antibodies, such smart thermally responsive MIP-NMs have the advantages of low costs, high stability, and so on. Moreover, thermally responsive switches of MIP-NMs show broad development prospects and application potential in biomedical science.

### 2.2. PH Responsiveness

The pH in the tumor microenvironment is quite different from that in normal organisms, so pH can be used as one of the environmental parameters of drug delivery systems. PH response refers to the corresponding change in response to changes in the external pH or ionic strength. The principle is that polymer molecules contain dissociated groups that are easy to hydrolyze and protonate, and these dissociated groups are affected by the change in external pH. When the pH changes, the ion concentration inside and outside the polymer will change, resulting in the fracture or formation of internal hydrogen bonds or covalent bonds in the polymer. Therefore, the pH response forms of MIPs are shown in the affinity of the binding sites, the volume of the MIPs, the degradation of the cross-linked polymer network, and the surface property of the MIPs. Because the main interaction between templates and functional monomers is hydrogen bonds, the external pH can affect most hydrogen bonds. Drug (template) release can be easily controlled by pH changes [95]. Polyacrylic acid (PAA) [96,97], polymethacrylic acid (PMAA), and other high-molecular-weight polymers are usually introduced for the preparation of pH-responsive MIP-NMs [98], because of the large number of ionizable carboxyl groups (-COOH) on their molecular chains. Qin et al. used N, N’-diacrylylcystamine (BAC) as the cross-linker and dimethylaminoethyl methacrylate (DMAEMA) as the main monomer to prepare DOX nanocarrier MIPs with an epitope of CD59 cell membrane glycoprotein (for cell targeting) as the template. The framework of MIP-NMs is broken under the stimulation of a tumor microenvironment, leading to the release of DOX through biodegradation [99]. In some other pH-responsive systems, DMAEMA was used to copolymerize with NIPAAm to modify the surface of MIPs with polymer brushes, in which the LCST of the brushes can be changed by the pH [93].

In recent years, borate affinity materials have attracted more and more attention, because boronic acid covalently interacts with cis-diol-containing molecules, forming covalent complexes at relatively high pH values, and dissociates reversibly in an acidic environment. The application of boronate affinity materials to molecular imprinting and the introduction of boronic acid functional monomers endow MIPs with many advantages, such as high affinity, reversible binding, and high tolerance [100]. Xie et al. combined the advantages of borate affinity and cationic pH responsiveness to propose pH-sensitive novel magnetic MIP-NPs for the selective separation of glycoproteins [101]. The MIP layer was composed of 4-vinylphenylbronic acid (VPBA), 2-(dimethylamino) ethyl methacrylate (DMA), and N, N-methylenebis acrylamide (MBA) via surface imprinting. The results showed that the MIP-NPs exhibited swelling or shrinking behaviors at different pH values, and the adsorption performance showed better adsorption capacity and elution efficiency than that prepared by traditional molecular imprinting. Recently, Chen et al. proposed a pH-responsive hollow MIP-NP preparation method based on surface imprinting and borate affinity for the selective separation of ribavirin in water samples [102]. Silica nanoparticles were used as sacrificial carriers to fix ribavirin on the surface of silica, and then monomer 3-acrylamide phenylboric acid (AAPBA), AAM, and cross-linking agent Bis were introduced to form an imprinted layer under the action of an initiator. Finally, the silica matrix and the template were removed using hydrofluoric acid and an HAc solution to obtain hollow MIP-NPs. The results show that the hollow MIP-NPs have high selectivity and enrichment for ribavirin and can be used to separate ribavirin from complex water samples. Due to the covalent interaction between the cis-diol structure of ribavirin and the phenylboric acid dissociated under acidic conditions, the pH value has a certain influence on the adsorption of the hollow MIP-NPs. The amount of ribavirin absorbed into the hollow MIP-NPs increases with the increase in pH value. Moreover, the pH-sensitive property makes the adsorption process reversible, so the hollow MIP-NPs can be reused.

### 2.3. Light Responsiveness

The light response method is also one of the more common responsive stimulation methods because of its easy and precise control, safety, and environmental friendliness. Under the irradiation of light sources, such as ultraviolet light, visible light, or infrared light, the light-responsive material in the photosensitive group receives the light signal, and the light signal is converted into a chemical signal through a light–chemical reaction, which changes the physical and chemical properties of the material [103]. Light-responsive MIPs usually introduce light-sensitive monomers to achieve reversible light-controllable smart responses. Common light-sensitive stimuli groups mainly include azobenzene (AZO), spiropyran (SP), diarylethene (DTE), and so on [104,105,106].

Azobenzene is the most common and most widely used light-sensitive materials, with both cis and trans isomers. Under irradiation of 340–380 nm ultraviolet light, the trans form changes to the cis form, and when the light source is converted to visible light or under the action of heating, the cis-form returns to the trans-form. Because the change in cis–trans will not cause the breaking of chemical bonds, many studies are carried out based on the special properties of azobenzene. For example, Liu et al. used azobenzene derivatives as functional monomers and the drug paracetamol (APAP) as a template to prepare NIR-light-responsive surface MIPs (NSMIPs) by atom transfer radical polymerization [107]. The photo-regulated release of APAP in aqueous solutions was induced by light irradiation. The substrate core selected inorganic nanoparticle UCNPs that can convert near-infrared (980 nm) to green light (520–550 nm). The green light is absorbed by the azobenzene-containing MIP layer on the surface of the UCNP, resulting in a cis–trans transition. Thereby, the drug release is induced, and the application example of NSMIP in the field of drug delivery is realized for the first time. Hanieh et al. prepared novel light-responsive MIP-NP-conjugated hyperbranched polymers for the selective recognition and intelligent separation of ATP from complex samples [108]. They developed light-responsive MIP-NPs using a water-soluble azobenzene derivative containing two hydroxyl groups (-OH), 5-[(4,3-(methacryloyloxy) phenyl) diazenyl] dihydroxy aniline (MAPDHA) as the functional monomer and EGDMA as the cross-linker. In pharmaceuticals and biological fluids, ATP release and uptake can be controlled under lights at 365 nm and 440 nm, respectively.

### 2.4. Gas Responsiveness

The above-mentioned stimulus responsiveness is usually triggered by strong stimuli, such as obvious fluctuations in temperature, significant changes in pH value, and strong light radiation, which may cause damage to the imprinted layer and reduce the sensitivity of MIPs. From the perspective of biocompatibility and toxicity in biological applications, this newly developed stimuli response mode of gas response is undoubtedly a good choice. Kitayama et al. first developed gas-sensitive MIP-NPs that used CO_2_ and N_2_ as gas stimuli to recognize the target molecule human serum albumin (HSA) in responses [109]. Using the gas-responsive initiator 2,2′-azobis [2-(2-imidazolin-2-yl) propane] (VA-061), which has an imidazoline group, as a functional initiator leaves the imidazoline groups in the binding cavities of the prepared MIP-NMs. Moreover, the groups react with CO_2_ dissolved in water to form bicarbonate, while the reverse reaction can be initiated when N_2_ is introduced. Introducing CO_2_ or N_2_ to convert the functional groups from a charged state to an uncharged state exhibits strong affinity and high selectivity for target proteins in CO_2_-treated aqueous media compared to those treated in N_2_.

Recently, Wang et al. introduced a CO_2_-sensitive functional monomer, DMAEMA, to construct a gas-responsive MIP-NP photochemical sensor. Zeolite imidazole framework-8 (ZIF-8) with a high specific surface area and the hepatitis virus (HBV) were used as the sacrificial material and template, respectively [110]. The cyclic adsorption and release of the template HBV were achieved by the intrinsic change in the amino groups in the imprinted cavity caused by N_2_/CO_2_ switching. The gas response has many advantages, not only is gas abundant, inexpensive, and non-toxic, but it also does not impair the response of materials due to salt accumulation compared to pH-responsive materials. Moreover, carbon dioxide is one of the normal metabolites of organisms and has biocompatibility, so molecular recognition materials responsive to gas stimuli can be applied to various biological applications, such as reversible affinity chromatography, reusable sensors, cell adhesion control functional substrates, and analytical tools for cells to identify metabolic disorders.

### 2.5. Reduction Responsiveness

In addition, great progress has been made in the development of reductive responses. The use of reduction-responsive stimuli for drug delivery and drug release has attracted more and more attention due to their efficient and rapid release. Polymeric drug delivery systems sensitive to the reduction process are often designed to respond to internal microenvironmental stimuli by exploiting the high gradient levels of glutathione (GSH) present in some types of cancer. Reduction-responsive MIP-NMs have been explored and are often designed from polymers containing disulfide bonds. The disulfide bonds in MIPs are rapidly cleaved in the presence of reducing agents, especially intracellularly in tumor cells [111].

Such reduction responsiveness systems mainly use two different strategies: the insertion of disulfide bonds in the polymer backbone or the use of reduction-sensitive cross-linking molecules [112], which can be incorporated into the core or shell of appropriate nanocarriers. It is worth mentioning that Zhao et al. published an article on reduction-responsive MIP-NGs [113]. They used the β-blocker S-propranolol for cardiac disease as a template molecule, and BIS(2-methacryloyloxyethyl) disulfide (DSDMA) or N, N-bis(acryloyl) cystamine (BAC) as the disulfide-containing cross-linker to synthesize reduction-responsive MIP-NGs, which exhibited better binding properties for S-propranolol than that constructed with conventional cross-linker EDMA (Figure 8). Moreover, in the presence of intracellular concentrations of GSH, the rate of S-propranolol release was higher, suggesting that these materials could potentially be used as intracellular controlled drug delivery systems.

### 2.6. Solvent Responsiveness

Solvent responsiveness is a simple and versatile stimuli-responsive method, which will play an important role in the design and preparation of smart stimuli-responsive materials in the future. Solvents play a crucial role in determining the binding properties and selectivity of MIPs and are easily driven by volatile solvents, such as water, acetone, and acetic acid [114]. Most MIPs are prepared in non-polar solvents under H-bond interactions between a polymer and template. MIPs shrink or swell when they are exposed to different organic solvents, which deforms their selective binding sites and reduces their ability to recognize template molecules [115]. Therefore, it is possible to make full use of the characteristics of the solvent and select the appropriate solvent to give the MIPs an excellent response performance and achieve the purpose of improving the separation efficiency.

Maddalena et al. prepared solvent-responsive MIP-NGs for the targeted recognition of human serum transferrin (HTR) from a serum [116]. The neutral acrylamide, hydrophobic tert-butylacrylamide, and charged itaconic acid monomers were mixed in the presence of the epitope template CGLVPVLAENYNK (CK13) to synthesize poly (acrylamide)-derivative PAD-MIP-NGs with responsiveness to acetonitrile solvent stimulation. Upon the addition of acetonitrile, the instantaneous shrinkage of the MIP-NGs was observed at all the acetonitrile concentrations tested. Ratios of acetonitrile/water that were 1:1 and 7:3 (*v*/*v*) caused the PAD-MIP-NGs to shrink to about one-third of their initial size, and then partially reswell to approximately 80% of the initial size, reaching the solvent diffusion equilibrium in the MIP-NG network. Moreover, the compatibility and potential use of MALDI-TOF-MS were analyzed, which will provide new perspectives and new ideas for targeted protein analyses.

### 2.7. Dual/Multiple Responsiveness

Dual or multi-responsiveness, as the name implies, refers to responses to multiple external stimuli, such as thermal/light, thermal/pH, magnetism/pH, magnetism/light, magnetism/thermal, and thermal/pH/magnetism double/multiple responses. Compared with single-response MIPs, it has multifunctionality and a strong responsiveness, and the development of MIP-NMs with a multi-stimuli response and multi-target targeting is key to future research.

Zhang et al. prepared a pH/GSH dual-responsive MIP-NP drug delivery system, revealing that the MIP-NPs have obvious pH/GSH responsive properties and can promote specific drug release in target cells through microenvironmental stimulation [117]. Lin et al. prepared multi-responsive magnetic core/shell-imprinted nanospheres as drug carriers to achieve the intelligently controlled release of ibuprofen under the stimulation of temperature and UV light [118]. Xie et al. developed dual-responsive MIP-NPs with both optical and magnetic stimuli [119]. The results showed that the dual-responsive MIP-NPs exhibited an excellent and rapid separation of their magnetic properties, and at the same time, they also exhibited good photo-responsive selectivity to the template protein bovine hemoglobin (BHb) under alternating light irradiation of 365 nm and 440 nm. Therefore, it can be seen that, combined with light-controlled release and uptake, rapid magnetic separation, and specific recognition, the resulting MIP-NPs with both light and magnetic responsiveness show significant advantages in the enrichment of trace proteins in biological samples. In addition, Fan et al. prepared novel, magnetic, hollow, molecularly imprinted nanospheres responsive to multiple stimuli (magnetic/thermal/pH), which showed good recognition ability and high adsorption to bovine serum albumin [120]. Some other MIP-NMs with multi-responsive properties were also reported, which can be found in other works in the literature [121,122,123,124].

All the above stimuli responses are derived from the polymer network of MIPs, such as the swelling and shrinkage of functional skeletons, or the deformation of imprinted sites under stimuli. Moreover, another kind of response is caused by synergies of other composite materials with MIPs. For example, magnetic Fe_3_O_4_ nanoparticles as the composite material result in a magnetic response on MIPs. Luminous units can be used as the core of core/shell MIP-NPs, which can display different signals after the templates are bound. These responses are common in biomedical applications, such as bioanalysis, bioimaging, tracking in vivo, and so on. We will introduce them in detail in the following part.

## 3. Biomedical Applications of Stimuli-Responsive MIP-NMs

The development of SR-MIP-NMs has attracted more and more attention in the research community. It is used in biomedicine effectively, including in bioimaging, disease diagnosis, drug delivery, disease intervention, etc. [125], which will be introduced in detail below.

### 3.1. Bioanalysis and Diagnosis

SR-MIP-NMs are often used as biosensors in the applications of bioanalysis and diagnosis. As we know, biosensors are required to have intuitive signal changes, but traditional MIPs must be compounded with other materials that can provide signals to meet this requirement. Therefore, these kinds of composite MIPs will produce different signals before and after binding template molecules, which can be also regarded as stimuli-responsive MIPs.

According to statistics, many people die of cancer every year [126,127,128]. Many cancers do not have obvious symptoms at the early stage and are always diagnosed at the terminal stage by conventional methods. Therefore, the specific identification, detection, and early diagnosis of cancer cells in the human body are of great significance for reducing mortality and prolonging the human lifespan. The current disease detection methods are mainly based on immunoassays, such as enzyme-linked immunoassays, radioimmunoassays, and fluorescent immunoassays. These methods often have problems, such as cumbersome operations, radioactive pollution, and many influencing factors, which limit the process of early disease monitoring, and searching for alternative new methods has become a research hotspot. Because the related signals can respond to the affinity of templates, SR- MIPs-based sensors have become promising analytical devices, which are used to detect kinds of cancer markers to diagnose cancer [129]. For example, lysozyme (Lyz) is considered a cancer marker, and its detection is essential for the early diagnosis of cancers. Wang et al. synthesized silica cross-linked MIP-NPs by the sol-gel method, which was applied for constructing selective fluorescent sensors to determine Lyz (Figure 9a) [130]. Finally, the constructed MIP@CdTe QD nanoparticles were successfully used to detect Lyz in human serum and chicken egg white, giving recoveries ranging from 95.6% to 99.2%. Such MIP-NPs exhibit huge potential in the field of the tearly diagnosis of cancer makers. Similarly, hexanal is a biomarker of lung cancer and can be detected in exhaled breath. Mousazadeh et al. used hexanal imprinted MIP-NPs and gold nanoparticles to construct a chemiresistive sensor by drop casting on the surface of an interdigitated electrode (Figure 9b) [131]. The electrical response and resistance change in the nanocomposite sensor can be detected when the hexanal gas is transferred to the test chamber by using the syringe. Moreover, the sensing layer showed good selectivity for hexanal rather than other interfering analytes with the detection window in the ppm range. This sensor showed the potential to detect hexanal gas in the headspace of biological matrices, such as cell culture medium, serum, plasma, urine, and saliva.

Because of their specific selection and recognition performance, low costs, and simple and fast operation, MIPs are used to screen and diagnose diseases with high morbidity and high mortality that threaten human health. For example, mycobacterium leprae is a causative bacterium that causes leprosy, and if left untreated, it can become a chronic disease. Archana et al. developed an electrochemical quartz crystal microbalance sensor modified with MIP-NPs for imprinting bacterial epitopes [132]. The MIP-NPs were electropolymerized on gold-plated quartz electrodes, and the sensor was able to show specific binding to mycobacterium leprae in the blood samples from infected patients. Therefore, MIP-NMs are effective diagnostic tools for bacterial diseases, which will expand clinical possibilities and enable effective population screening.

The development of smart stimuli-responsive nanomaterials is one of the main driving forces of nanomedicine, so MIP-NPs have been widely studied as a novel targeting ligand for virus detection [133]. There have been many types of viruses identified based on MIPs, such as bacteriophage, adenovirus, dengue virus, influenza A virus, pneumonia virus A, norovirus, and so on. Japanese encephalitis virus (JEV) is one of the main pathogenic factors of encephalitis. At present, the early and rapid detection and diagnosis of JEV are usually carried out by collecting patient serum for detection tests, such as hemagglutination inhibition tests, complement fixation tests, spot enzyme immunoassays, etc. However, these techniques are time-consuming and complex or require species-specific reagents. Luo et al. prepared a novel, magnetic, molecularly imprinted resonance light-scattering (MIPs-RLS) sensor (Figure 9c) [134]. They chose to imprint on the surface of Fe_3_O_4_@SiO_2_, and the functional monomer aminopropyl-triethoxysilane (APTES) was utilized for the fixing target molecule JEV through polymerization with tetraethyl-orthosilicate (TEOS). The MIP-NPs used magnetic silicon microspheres acting as carrier materials that enabled the imprinting microspheres’ fast magnetic separation. The readily separated JEV sensor prepared in this work had a detection limit of 1.3 pM and could reach an absorption equilibrium within 20 min. Combining the advantages of RLS and MIPs, it not only shows a low detection limit but also a high selectivity. In addition, the cost is lower, no expensive equipment is required, and it is easy to operate. Therefore, this method can be used as an effective means of virus detection and diagnosis. In addition, Cai’s research group has been committed to intelligent responsive MIP virus detection. In 2017, the thermal-response MIP-NPs for virus detection were first proposed and implemented [135]. A thermal-sensitive MIP resonance light-scattering (RLS) sensor based on hepatitis A virus (HAV) detection was prepared. By introducing NIPAAm to polymerize on the surface of SiO_2_, the as-prepared thermally responsive HAV-MIP-NPs could achieve identification and detection in as fast as 30 min, and the recognition performance of the target virus could be adjusted by controlling the temperature. A MOF-based pH-responsive RLS MIP-NP assay was subsequently proposed for the detection of HAV [136]. A dual-response MIP sensor was developed in 2021 [137], based on light/magnetic responses, in order to recognize enterovirus 71 (EV-71). Recently, based on the strong interest in virus detection, a CO_2_ gas-responsive MIP sensor has been developed, which can accurately identify the hepatitis B virus (HBV) (Figure 9d) [110]. The gas response sensor has excellent stability and reproducibility and is easy to recycle. This strategy is expected to provide a new green method for the simple and rapid screening of viral infections as well as the control of viral epidemics.

**Figure 9 molecules-28-00918-f009:**
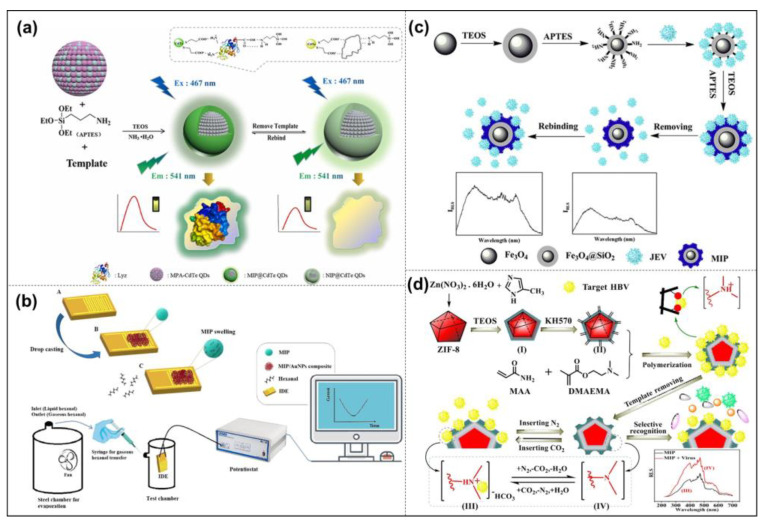
Schematic diagram of the synthesis of MPA-CdTe QD-embedded MIPs (**a**); a conductive nanocomposite of MIPs and AuNPs for detection of lung cancer (**b**); synthesis of magnetic Fe_3_O_4_ nanoparticles, silica-modified Fe_3_O_4_ nanoparticles, functionalized Fe_3_O_4_@SiO_2_ microspheres, virus-imprinted magnetic microspheres, and non-imprinted microspheres (**c**); preparation of MIP-NPs and principle of HBV recognition (**d**). Reprinted with permission from Refs. [110,130,131,134]. Copyright 2020, Elsevier; 2022, Elsevier; 2019, Elsevier; 2022, Elsevier.

In recent years, great progress has been made in the imprinting of target templates in the nanoscale size range, especially relying on surface imprinting to improve binding capacity. MIP-NMs, a new detection and diagnosis technology, are very important to improve disease treatment, and as biomimetic identification materials, they have a high value in practical applications. Their selectivity, sensitivity, stability, and reproducibility give MIP-NMs a great advantage in diagnostic research. Nevertheless, there are still certain challenges in this field, such as the inherent heterogeneity of virus template molecules, the difficulty in obtaining pure template molecules, and the low reproducibility of large-scale productions.

### 3.2. Bioimaging

Similar to the above fields of biological applications, the luminescent properties of MIP-NMs come from fluorescence molecules, quantum dots, carbon dots, etc. However, some of these luminous signals are not changed after the templates are bound, which are different from those of MIPs’ biosensors. This kind of SR-MIP-NM provides a means of tracking MIPs.

Bioimaging technology is used to locate target ligands in tissues and cells inside organisms. Biological information is obtained by the qualitative or quantitative determination of biomolecules and is expressed in the form of images to implement the early detection and diagnosis of diseases, surgical guidance, treatment effect tracking, and prognoses. Bioimaging is also an important tool in the fight against cancer [12]. Due to the advantages of fluorescence imaging, such as its low costs, high resolution, and the number of molecules that can be detected, fluorescence detection technology has become the most commonly used method in the field of biological analysis. Traditional fluorescent labels such as organic dyes are often unstable, prone to photobleaching, and exhibit narrow absorption and broad emission spectra, resulting in a low detection sensitivity. Fluorescent nanoparticles make up for these defects and can achieve a strong fluorescent signal and greatly improve sensitivity. Compared with conventional natural biological recognition elements (such as fluorescent proteins, antibodies, enzymes, or aptamers), the “custom” synthetic materials of MIP-NPs have some advantages, such as good binding properties, reproducibility, no harm to animals, and low costs. In the imaging field, fluorescent labeling visualization technology is generally used in combination with MIP-NPs. For example, Shinde et al. prepared fluorescent core/shell MIP-NPs by introducing nitrobenzoxadiazole (NBD) fluorophores and selectively labeled cell surface glycans by imprinting sialic acid to achieve bioimaging (Figure 10a) [138]. The results showed such MIP-NPs selectively stained different cell lines in correlation with the SA expression level.

In the field of MIP-NP fluorescence imaging, the selection of efficient detection probes is very important [139]. Suitable luminescent imaging components, such as quantum dots (QDs) [140,141,142], carbon dots (CDs) [143], organic dye-doped polymers or silica NPs [144,145], conjugated polymers (CPs) [146,147], etc., and non-luminescent imaging modalities, such as gold NPs [148], silver NPs [149], radioisotope-enriched NPs and iron oxide NPs, are chosen to achieve different goals. Introductions to related materials can be found in other works in the literature.

QDs [150] and CDs [151] are the most common fluorescent response groups, which are always combined with MIPs in the field of cell imaging. For example, Maria et al. developed MIP-coated QDs for the simultaneous multiplexed labeling of human keratinocytes with green InP QDs conjugated with MIP-GlcA (glucuronic acid) and red ZnS QDs conjugated with MIP-NANA (N-acetylneuraminic acid) [140]. A confocal microscope image showed the simultaneous multiplexed staining of GlcA and NANA on human keratinocytes by both colors (Figure 10b). Their work demonstrates for the first time the potential of molecularly imprinted polymers when conjugated to quantum dots of different emission colors as a versatile multiplexed imaging tool. Demir et al. reported an article in 2018 that combined CDs with MIPs for cell-targeted bioimaging [152]. Using starch as the carbon source and L-tryptophan as the nitrogen atom donor, they prepared N-doped CDs by a hydrothermal synthesis, which produced strong fluorescence at 450 nm when stimulated with a UV light. Then glucuronic acid (GlcA)-imprinted MIP shells were synthesized on the CDs’ surface. Hyaluronic acid is a biomarker of some cancers (cervical cancer cells, breast cancer cells), and GlcA is an epitope of hyaluronic acid. Therefore, this nucleocapsid structure biologically targets the hyaluronic acid overexpressed on the surface of cancer cells to achieve biological imaging. However, the fluorescent signals originated from fluorescent dyes and semiconductor quantum dots, which may increase the toxicity or pose challenges for long-term monitoring. Recently, Zhang’s group used different fluorescent modules, silicon nanoparticles (Si NPs), and carbon dots (CDs) as their MIP-NPs’ cores to achieve targeting cell imaging, which has a low toxicity, stable fluorescence properties, and good biocompatibility. For example, they established dual-template MIP-layer-coated fluorescent Si NPs by using the linear peptide of the extracellular region of human epidermal growth factor receptor-2 (HER2) and DOX as the template for targeted imaging and targeted therapy (Figure 10c) [83]. Moreover, they used silica nanoparticles embedded with carbon dots as carriers and the linear peptide of the extracellular region of human epidermal EGFR as templates to synthesize oriented fluorescent imprinted polymer nanoparticles for the targeted imaging of cervical cancer by specifically recognizing the EGFR overexpressed on tumor cells. The cell imaging results in tumor-bearing mice show that MIP-NPs have a better imaging effect than non-imprinted particles, which further proves the capability of targeted imaging in vivo (Figure 10d) [153].

**Figure 10 molecules-28-00918-f010:**
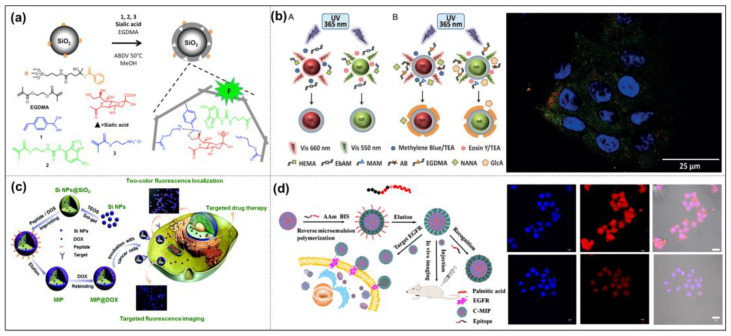
Procedure for SA-imprinted shell on silica core particles by a mixed covalent and noncovalent approach (**a**); red or green light emitted from InP/ZnS quantum dots excited by irradiation is used to synthesize a polymeric shell in situ around the particles by photopolymerization and the second shell of MIP is synthesized. Confocal microscope image showing simultaneous multiplexed staining of GlcA and NANA by MIPGlcA-QDs (green) and MIPNANA-QDs (red) (**b**); illustration for preparation of MIPs for targeted fluorescence imaging and targeted therapy in the cancer cell (**c**); preparation and application diagram of directed fluorescent MIP-NPs for cervical cancer targeted imaging and confocal fluorescence imaging of HeLa cells incubated with MIP-NPs and NIP-NPs 4h, respectively (**d**). Reprinted with permission from Refs. [83,138,140,153]. Copyright 2015, ACS; 2016, Wiley; 2019, Royal soc of chemistry; 2020, Springer.

Although bioimaging technology has broad prospects for development, the research on smart MIP cell imaging is still in its infancy. If it is to be practically applied in biomedicine, researchers still need to make further efforts, and there is still a long way to go in terms of research. In addition, most of the fluorescent signals are also in response to excitation; however, the cell imaging related to a template’s recognition properties is always displayed with the targeted drug delivery.

### 3.3. Drug Delivery

The responses of common SR-MIP-NMs are mainly reflected in the applications of drug delivery. The skeletons of SR-MIP-NMs shrink, swell, or biodegrade when their microenvironment changes, and drugs can be held stably or released fast.

In recent years, disease treatment through controlled drug release or drug-sustained release has been widely studied. Delaying or controlling drug release in the body not only reduces the frequency and side effects of drug use but also improves drug efficacy and safety. The realization of drug-controlled release and drug sustained-release must require the drug carrier material to have good biocompatibility and biodegradability. Functionalized MIP-NMs meet this requirement and can be combined with almost any drug molecule with a high affinity. MIP-NMs have a certain selective adsorption capacity towards template molecules, and their surface-imprinted cavities also have a high loading capacity. Therefore, MIP-NMs are used as effective materials for drug delivery systems in vivo, and their medical effects can be applied to the whole body. In 1998, Norell et al. proposed the idea of using MIPs to achieve sustained-release drug delivery [154]. They used methacrylic acid as a functional monomer to prepare theophylline-imprinted polymers and the result suggested that the specific binding properties of MIPs provide good prospects for the preparation of novel controlled-release drug systems. In recent years, the application of MIPs to drug delivery systems has become increasingly mature. The imprinting sites are not only used as drug carriers but also as recognition points of targeted molecules, because of their selectivities towards templates. In 2018, Canfarotta et al. prepared double-imprinted nanoparticles using the solid-phase imprinting method [155]. The epitope peptide of EGFR was used as the primary template to be immobilized on glass beads, and the doxorubicin drug molecules were mixed as the secondary templates when the monomers were added and mixed. In this way, MIP-NPs loaded with drug molecules were obtained, which were surface-imprinted against the epitope EGFR for accurate cell targeting while possessing binding sites within their bulk for drug delivery. Figure 11 left shows that MIP-NPs with surface cavities can recognize the targeted EGFR, and inner sites can hold the drug doxorubicin. Moreover, due to the EGFR targeting, the drug retention in the MDA-MB-468 cell is higher than that in the SKBR-3 cell after 24 h (Figure 11 right 5 vs. 2), while the drug retention in MIP-NPs is also higher than that in the pure DOX system (Figure 11 right 5 vs. 6). Such MIP-NPs can be considered as a new plausible alternative to conventional antibodies as both imaging and therapeutic tools against various clinically relevant targets.

In addition, many researchers have developed stimuli-responsive drug delivery systems inspired by the intelligent response of MIPs through the perception of changes in the external environment, triggering release and self-feedback drug delivery. Liu’s group developed double-imprinted smart MIP-NPs for specific targeting, prolonged retention time, and tumor microenvironment-triggered release [156]. The pH-responsive MIP-NPs were prepared by borate affinity-controlled orientation surface imprinting using 5′-Deoxy-5-fluorocytidine (DFCR) and SA as the tumor-targeted prodrug and marker, respectively. The microenvironment of the tumor site was slightly acidic, and the pH was about 6.5–6.8, which is unfavorable to the binding between DFCR and boric acid. Therefore, the MIP-NPs concentrated around the tumor cell through the affinity between the SA-imprinted cavities and the overexpressed SA on the tumor cell, and released the prodrug DFCR. Moreover, the drug release kinetics at different pHs showed that the DFCR would not be released from the imprinted cavities under the microenvironment pH (7.4) in normal tissues. This fully demonstrates that stimuli-responsive MIP-NPs not only exhibit the ability to specifically target tumor sites but also have great potential for prolonged retention and pH-triggered gradual release in tumor microenvironments (Figure 12a). Another study of the same group reported redox-responsive MIP-NPs for the targeted delivery of proteins for cancer therapy [81]. SA was imprinted on the surface of biodegradable silica nanoparticles, and ribonuclease A (RNase A) was encapsulated in the matrix of disulfide-hybridized silica NPs. Under the stimulation of the redox reaction, the S-S bond in the silica core was broken, resulting in the collapse and degradation of the MIP-NPs’ structure, and then RNase A was released into the body. Such nanovectors could not only maintain a high stability under physiological conditions but also permit redox-triggered biodegradation for both the concomitant release of the loaded therapeutic cargo and in vivo clearance. Such work will be able to serve as a potent platform for the targeted intracellular delivery of therapeutic proteins toward cancer therapy (Figure 12b). Both of their studies have shown obvious effects on tumor treatment.

In addition, simple capsule-like MIP-NPs for targeted and chemo-photothermal synergistic cancer therapy were constructed by Liu et al. In their design, dopamine (DA) was used as the functional monomer, cross-linking agent, and photothermal agent for the epitope EGFR imprinting. The PDA-based MIP layer was coated on the surface of a zeoliticimidazolate framework-8 with DOX loading [157]. MIP-NPs exhibit obvious photothermal transformation, excellent biocompatibility, and a controlled pH response to DOX release (Figure 12c). Moreover, cancer cells with high EGFR expression levels showed excellent targeting performance and good release performance, while normal cells showed low drug release. It is worth mentioning that such MIP-NMs in the above drug delivery systems also showed fluorescence imaging properties because of the fluorescent modules (fluorescein isothiocyanate isomer FITC-doped silica nanoparticles or fluorescent DOX).

In conclusion, MIP-NMs for stimuli-responsive drug delivery will become a promising multifunctional drug delivery carrier material. Because SR-MIP-NMs can selectively bind to template molecules, carry drugs, and realize controlled or sustained drug release by adjusting external environmental stimuli, they are expected to be applied in clinical medicine.

### 3.4. Disease Intervention

Disease intervention is different from clinical treatment. It refers to preventive treatment in advance of the inducements and aggravating factors of the disease, and it is a measure to prevent high-risk diseases. Early intervention for the disease and interventional disease treatment are very important. The prognosis of many diseases is directly related to the timing of treatments and the degree of occurrence of the disease, so timely identification and intervention measures should be taken. If the optimal treatment timing or effective time is missed in major diseases, it may lead to poor prognoses, death, and/or disability. Therefore, it is particularly important to intervene in the early stage of the disease, which can control the damage to a certain extent and reduce the risk. MIP-NMs provide an effective means for disease intervention. Unlike chemotherapy, MIP-NMs such as antibodies or inhibitors, especially interact with bioactive molecules and inhibit their biological activities with any chemical drugs. Liu’s group developed a new strategy utilizing MIP-NPs to inhibit HER2+ breast cancer growth without any drugs [158]. MIP-NPs were prepared by borate-affinity-controlled directional surface printing, which could bind almost all HER2 glycans, prevent the dimerization of HER2 with other EGFR members, block downstream signaling pathways, and thus inhibit the growth of HER2+ breast cancer. This study not only provides a new strategy for the treatment of HER2+ breast cancer but also suggests the potential of MIP-NMs in disease intervention.

In addition, Haupt et al. developed a series of PNIPAAm-based MIP-NGs via solid-phase synthesis to inhibit cell–cell interactions. For example, they used the temperature responsiveness of NIPAAm to prepare MIP-NGs of the epitope peptide SWSNKS (3S) for specific binding to the envelope glycoprotein 41 (gp41) of human immunodeficiency virus type 1 (HIV-1) [159]. The epitope peptide SWSNKS (3S) is associated with a decline in CD4 T cells and contributes to the deterioration of the immune system during HIV infection. The MIP-NGs were able to target and block the 3S peptide to prevent subsequent cascade interactions against CD4 T cells (Figure 13a). They also synthesized MIP-NGs using Asp1-Trp2-Val3-Ile4-Pro5-Pro6-Ile7 as epitope peptides to specifically recognize cadherin [160]. The cell aggregation test showed the addition of MIP-NGs results in dispersing cells, indicating an effect on the cell–cell adhesion, whereas non-imprinting NGs did not affect adhesion. Moreover, the MIP-NGs incubated together with the free template peptide could block the binding sites and did not affect the cell aggregation (Figure 13b). The results showed that MIP-NGs are considered as a promising therapeutic strategy for cadherin-mediated cancer, because the affinity of MIP-NGs towards the template peptides that are responsible for cancer cell–cell adhesion is more effective than that of commercially available antibodies in inhibiting cell adhesion assays, destroying 3D tumor spheroids, and inhibiting human cervical adenocarcinoma (HeLa) cell invasion. Moreover, since the tumor microenvironment has a denser extracellular matrix than normal tissues, it is difficult for drugs to pass through the tightly packed tumor cells. However, the MIP-NGs that effectively inhibit the adhesion between cells can relax these cells for drug penetration, so they have great potential in tumor therapy. Recently, their group used the same method to produce MIP-NGs based on the amino acid sequence His-Ala-Val (HAV) in the extracellular domain of N-cadherin, which is involved in cadherin-mediated cell adhesion and migration, to inhibit cell adhesion and the migration of HeLa cells [161]. Although MIP-NMs are still in their infancy for disease intervention, these findings confirm the fact that they have a legitimate place in this field, and various MIP-NMs are being developed to facilitate the effect of disease intervention. For example, the epitope peptide of a-synuclein (SNCA) was imprinted to prepare MIP-NPs, which can extract SNCA in Parkinson’s brain organoid culture medium. Such MIP-NPs have therapeutic and diagnostic potential for treating Parkinson’s [71]. 

Moreover, MIP-NPs designed by Teixeira et al. and templated with transforming growth factor-β3 (TGF-β3) could recognize and preferentially bind TGF-β3 in a complex human platelet lysate. The MIP-NPs were used for the 2D culture of adipose-derived stem cells, which suggested the activation of TGF-β3 signaling pathways without requiring any supplementation with exogenous GF. Such results proved that MIP-NPs can be used as potential alternative abiotic GF ligands to guide cell fate in tissue engineering and regenerative medicine applications [45].

From the development of MIP-NMs in biomedicine, we find that MIP-NMs are closer to the function of biological antibodies, especially in disease intervention. Whether in terms of their material structure or their physical and chemical properties, MIPs prepared via solid-phase synthesis are undoubtedly the best candidates. These MIP-NMs with thermal responsiveness, we believe, are bound to have more breakthrough applications soon.

In order to better understand the classification of MIP-NMs in this review, we sum-marized the references according to different templates and applications, which are listed in Table 1.

## 4. Conclusions and Outlook

MIPs were initially designed as plastic antibodies mimicking natural antibodies. With the advent of nanomaterials, MIP-NMs have become closer to natural antibodies in their performance and scale. Fortunately, the development of smart MIP-NMs has successfully endowed MIPs with the ability to respond to external stimuli, just like an antibody with intelligence. Unlike natural antibodies, MIP-NMs’ excellent stability facilitates long-term preservation and storage, which further improves the possibility of practical application. In recent years, great progress has been made in SR-MIP-NMs, and their feasibility in the biomedical field has been successfully confirmed, laying a solid foundation for their practical application in biomedical clinics one day. Moreover, from a practical and economic point of view, it is not surprising that SR-MIP-NMs are socially acceptable in biomedical applications [162].

However, there are still some problems that we have to consider carefully before using SR-MIP-NMs as antibody substitutes. (1) Which part of the template is imprinted is very important. As we know, antigens or macro biomolecules have their special active sequences, which show their biological activities. SR-MIP-NMs should recognize these sequences, to inhibit the activities of the antigen or biomolecules. Therefore, fixing the orientation of the template is extremely important, whether it is a protein or an epitope peptide. For example, the relative bioactive domain of a protein should be exposed to ensure that SR-MIP-NMs are formed and the domain is covered. Only the SR-MIP-NMs obtained in this way will have the imprinting sites with a highly efficient inhibition performance; on the contrary, although SR-MIP-NMs combine well with template protein, their activity cannot be inhibited. (2) The selection of raw materials and solvents during preparation has a great impact on a SR-MIP-NM’s performance. Functional monomers and template molecules should have a stronger binding force, which depends on the choice of monomers and solvents. Many solvents used in the synthesis of SR-MIP-NMs are organic solvents, which facilitate the dissolution of monomers as well as facilitate the interaction of the functional monomers and template molecules, but are not suitable for use in living organisms. Therefore, solvents for the synthesis of SR-MIP-NMs are required to be non-toxic to mimic the liquid in vivo, or the monomers and cross-linkers are required to be hydrophilic to enhance the binding force between SR-MIP-NMs and template molecules in aqueous solutions [163]. (3) What happens to SR-MIP-NMs after treatments in vivo should be considered further. Most studies showed SR-MIP-NMs are non-toxic; biocompatible polymers are friendly to cells. However, after disease treatments or interventions, are these SR-MIP-NMs directly excluded from the body? Or are they biodegradable? Some oligomers and small molecules produced by biodegradation may be harmful to the human body. Although the choice of raw materials is very important during preparation, related research on what happens after biodegradation is still rare, which is also an obstacle to SR-MIP-NMs’ practical applications. All in all, intelligent SR-MIP-NMs have broad application prospects in the field of biomedical science and will become a new development trend, but their development still faces many of the above problems, which need to be solved by researchers.

## Figures and Tables

**Figure 1 molecules-28-00918-f001:**
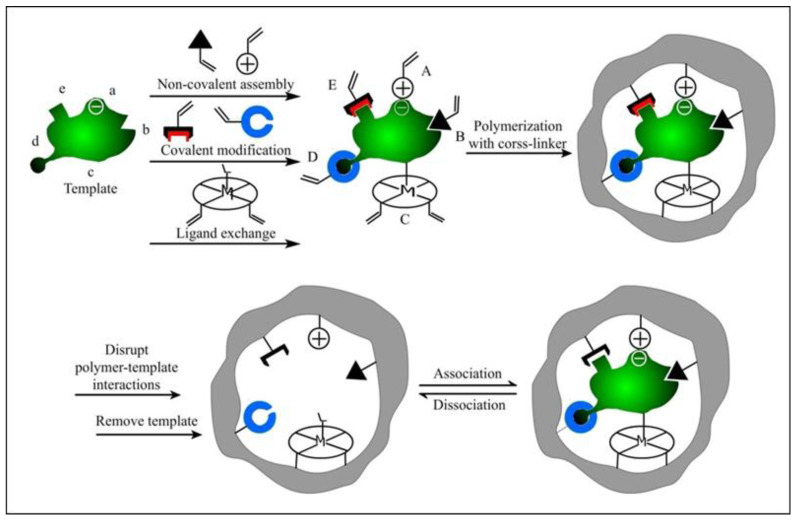
Schematic representation of the molecular imprinting process. Reprinted with permission from Ref. [13]. Copyright 2015, MDPI.

**Figure 2 molecules-28-00918-f002:**
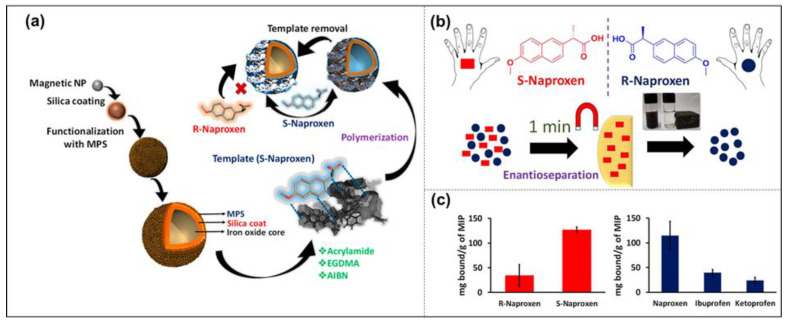
The overview of the synthesis of (S)-naproxen-imprinted nanoparticles (**a**); rapid magnetic separation of enantiomers (**b**); selective nature towards enantiomer, (R)-naproxen, and other similar structures, ibuprofen and ketoprofen (**c**) Reprinted with permission from Ref. [41]. Copyright 2019, ACS.

**Figure 3 molecules-28-00918-f003:**
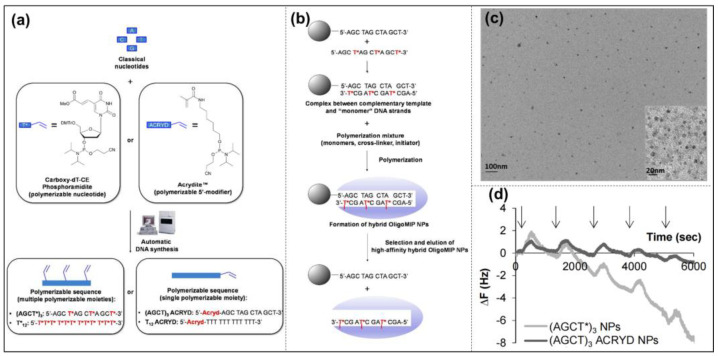
MIP−NPs were prepared using a solid-phase blot polymerization strategy using oligomer DNA sequences as template molecules. Schematic for the preparation of polymerizable oligonucleotide sequences used as recognition elements in the MIP−NPs composition (**a**); schematic representation of the solid-phase synthesis and selection of oligo MIP−NPs (**b**); TEM image of MIP−NPs (**c**); QCM graph of (AGCT)3 acrylic MIP-NPs and (AGCT*)3 MIP−NPs (**d**). Reprinted with permission from Ref. [59]. Copyright 2016, Royal soc of chemistry.

**Figure 4 molecules-28-00918-f004:**
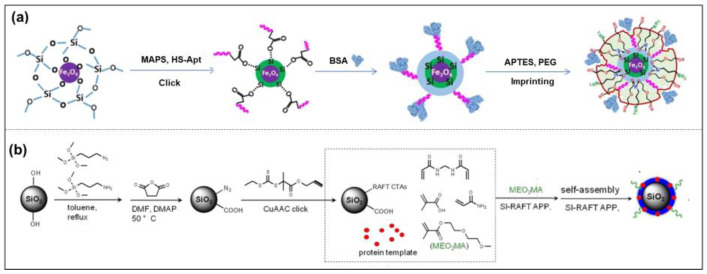
The two different methods for protein imprinting are prefixed (**a**), and unfixed (**b**). Reprinted with permission from Refs. [64,66]. Copyright 2018, Springer; 2019, Royal soc of chemistry.

**Figure 5 molecules-28-00918-f005:**
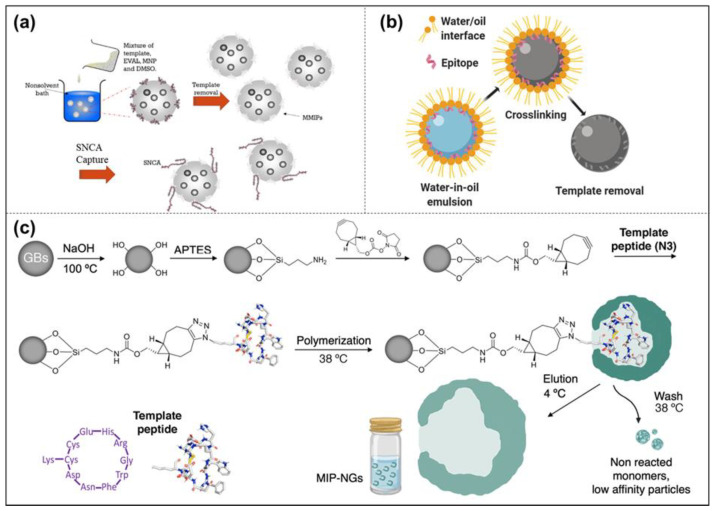
Methods of epitope imprinting. The protocols of epitope imprinting via precipitation polymerization (**a**) inverse emulsion polymerization (**b**); and solid-phase synthesis (**c**). Reprinted with permission from Refs. [45,70,71]. Copyright 2022, MDPI; 2020, Wiley; 2021, Wiley.

**Figure 6 molecules-28-00918-f006:**
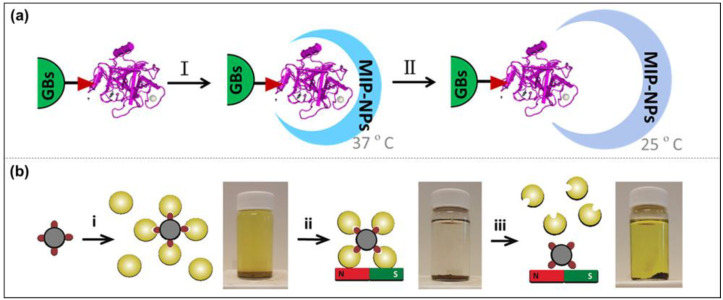
Schematic diagram of solid-phase synthesis of MIP-NPs on glass beads (**a**) and on magnetic beads (**b**). Reprinted with permission from Refs. [76,77]. Copyright 2016, ACS; 2019, Wiley.

**Figure 7 molecules-28-00918-f007:**
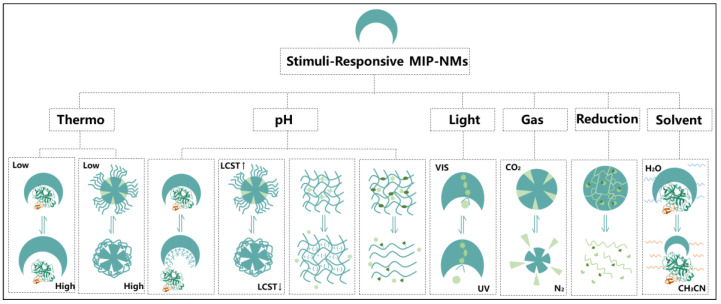
Schematic diagram of SR-MIPs with different stimulus responses.

**Figure 8 molecules-28-00918-f008:**
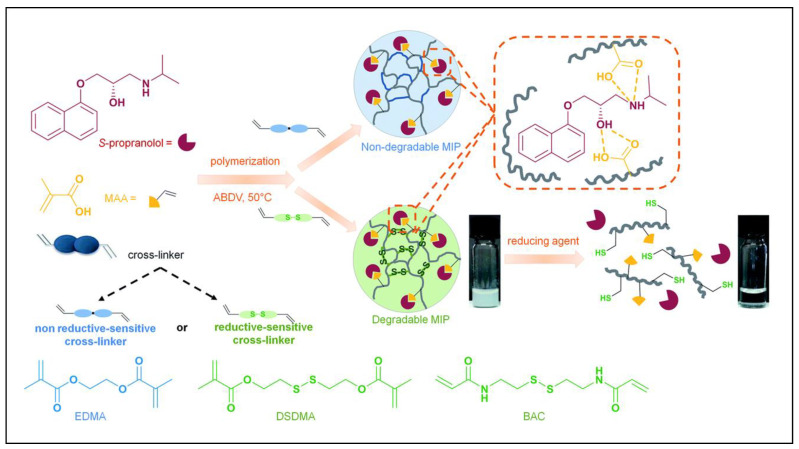
Schematic representation of syntheses and reduction-induced degradation of S-propranolol molecularly imprinted polymers for drug delivery. Reprinted with permission from Ref. [113]. Copyright 2020, Royal soc of chemistry.

**Figure 11 molecules-28-00918-f011:**
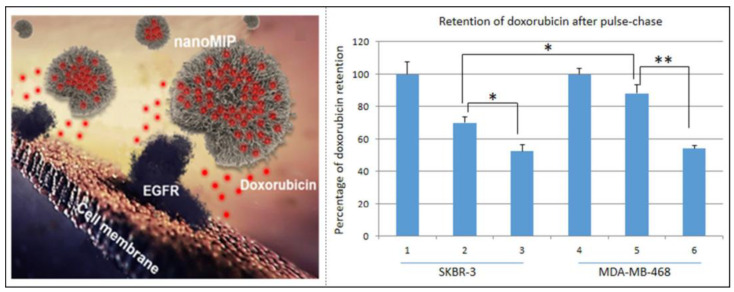
Double-imprinted MIP-NPs are loaded with adriamycin and target linear epitopes of EGFR (**left**); e retention of doxorubicin after pulse-chase in SKBR-3 and MDA-MB-468 cells. * *p* < 0.05, ** *p* < 0.01. (**right**). Reprinted with permission from Ref. [155]. Copyright 2018, ACS.

**Figure 12 molecules-28-00918-f012:**
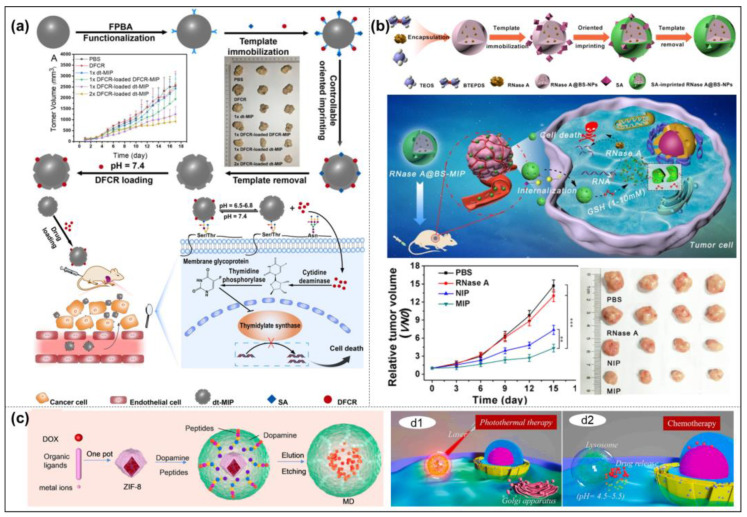
Schematic of the drug transport and action mechanism in the dual-templated MIP-based smart prodrug delivery system. Representative photographs of mice in different groups after treatment for 18 days and mean tumor volume (**a**); illustration of SA-imprinted MIP-NPs applied for transport in blood vessels, tumor targeting, cellular uptake, and sequential GSH-triggered degradation along with cargo release. ** *p* < 0.01, *** *p* < 0.001. (**b**); the detailed synthetic procedure of MIP-NPs, and chemo-photothermal synergistic anti-cancer therapy (**c**). Reprinted with permission from Refs. [81,156,157]. Copyright 2020, Wiley; 2021, ACS; 2021, Elsevier.

**Figure 13 molecules-28-00918-f013:**
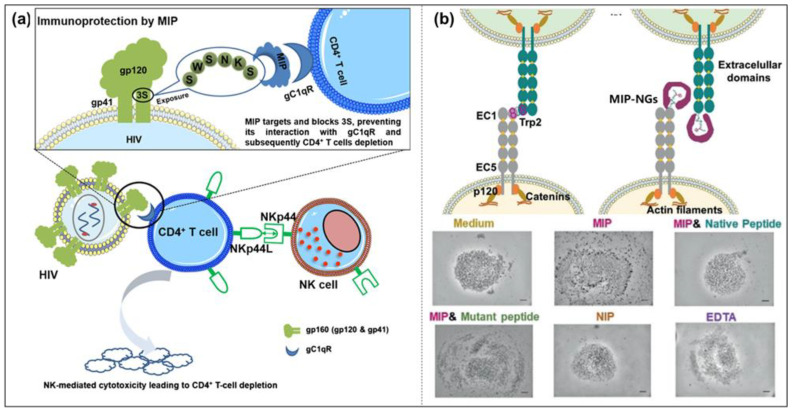
Mechanism of the action of MIP targeting and blocking 3S (**a**); MIP-NPs targeting an N-terminal epitope of cadherin, block Trp2 and abrogate adhesion between opposite cells, and cell aggregation of HaCaT cells when incubated with medium, MIP-NPs, MIP-NPs in the presence of the native peptide or the mutant peptide 2, NIP-NPs, and EDTA (**b**). Reprinted with permission from Refs. [159,160]. Copyright 2019, ACS; 2019, Wiley.

**Table 1 molecules-28-00918-t001:** Summary of references with different template molecules and corresponding applications.

The Template Molecules			References	Application	References
Small Molecule Imprinting	Drug Molecules	paracetamol	[41,47,83,99,102,107,113,117,118,155,156,157]	separating and purifying target	[41,102,107]
		(S)-naproxen			
		doxorubicin			
		ribavirin		drug delivery	[47,83,99,113,117,118,155,156,157]
		S-propranolol			
		ibuprofen			
	Small Biomolecules	short peptide	[42,43,44,45,46,47,48,69,70,71,116]	identifying the target	[42,43,46,47]
		monosaccharide		recognizing proteins	[44,45,48,69,70,71,116]
Biomacromolecule Imprinting	Polysaccharide	heparin	[53,94]	separation of function	[53]
		hyaluronic acid			
		fucoidan		detection of function	[94]
	Nucleic Acid	DNA or RNA	[58,59]	detection of genetic diseases	[58]
				identifying and detecting nucleic acid targets	[59]
	Protein	glycoproteins	[63,64,65,66,72,76,77,101,109,116,119,120,130]	separation and purification	[63,64,65,66,72,76,77,101,109,116,119,120,130]
		bovine hemoglobin			
		lysozyme			
		trypsin			
Cell Imprinting	Proteins and lipids on the surface of cell membranes	epidermal growth factor	[81,82,83,99,138,140,152,153,155,156,157,158,160,161]	identifying target cells	[81,82,160,161]
		glycoprotein		biological imaging	[83,138,140,152,153]
	Sugar structures on the surface of cell membranes	sialic acid		targeted cancer therapy	[83,99,155,156,157,158]
		B-type blood trisaccharide			
Others	microorganisms	bacteria	[110,132,133,135,136,137,159]	microbial detection	[132]
		fungi			
	virus	HBV		virus detection	[110,133,135,136,137,159]

## Data Availability

Not applicable.

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
