# Peer review of "Molecularly Imprinted Nanomaterials with Stimuli Responsiveness for Applications in Biomedicine"

_molecules, 2023, doi:10.3390/molecules28030918_

Round 1
Reviewer 1 Report
We acknowledge that this is a vigorous review, citing 162 publications. However, the paper is not well organized.
Only the second half of the paper discusses the subject of stimulus-response MIP. Also, in the second half of the paper, there are some papers (e.g., Ref. 154) that make it questionable whether the stimulus-response is effectively utilized. The first half should be shortened and the second half should clarify the principles of stimulus response and the benefits of introducing stimulus response in each study.In addition, among the stimulus responses of MIP, sensors that respond to specific interactions with the template itself as a stimulus have been vigorously developed recently. We recommend introducing such papers as well.
Thus, I recommend that the paper is rejected now and look forward to resubmission with the above modifications.
Author Response
Point: We acknowledge that this is a vigorous review, citing 162 publications. However, the paper is not well organized.
Only the second half of the paper discusses the subject of stimulus-response MIP. Also, in the second half of the paper, there are some papers (e.g., Ref. 154) that make it questionable whether the stimulus-response is effectively utilized. The first half should be shortened and the second half should clarify the principles of stimulus response and the benefits of introducing stimulus response in each study. In addition, among the stimulus responses of MIP, sensors that respond to specific interactions with the template itself as a stimulus have been vigorously developed recently. We recommend introducing such papers as well.
Thus, I recommend that the paper is rejected now and look forward to resubmission with the above modifications.
Author response: Thank you very much for your encouraging and very valuable comments. According to your suggestion, we have revised the first half of the paper and simplified some contents. The second half of the study was carefully checked, and on the basis of clarifying the principle of stimulus response, some studies that does not involve the benefits of stimulus response were supplemented. The revised parts were marked up using the “Track Changes” function in the revised manuscript.
Reviewer 2 Report
The manuscript is a review on molecularly imprinted nanomaterials (MIPs), with a focus on stimuli-repsonsive MIPs. Section 1 gives an introduction on this subject, section 2 describes stimuli-responsive MIPs categorized by response, and section 3 discusses the use of MIPS in biomedical applications. I find the review interesting and generally well structured, and think it is suitable for publication. I do have some comments that could help further improve the manuscript:
-There are various spelling and grammar mistakes throughout the manuscript. Also the figures are often of rather poor image quality.
-While the topic of the manuscript is on stimuli-responsive MIPs, and this subject is extensively discussed in section 2, the conclusions hardly mention stimuli-responsiveness. Instead, the conclusions section lists some general challenges in the field of MIPs. This could be improved.
-It is not always clear if the authors refer to polymers or polymer particles when they describe MIPs. When reading the beginning of section 1.2 I was initially confused if MIPs refers to dissolved polymers, polymer particles or bulk polymer materials. It does not help that the authors also use the word "bulk polymers" to describe the materials. This also leads to a weird situation in line 555 where it says "the ion concentration inside and outside the polymer will change", where I think "polymer particle" is intended.
-Line 94: "Since MIP-NMs have comparable selectivity and affinity for natural receptors...". I doubt that "receptors" is the correct word here.
-Line 126-128: Is the polymer on the magnetic nanoparticle a layer or particles? How is it bound to the nanoparticle?
-Line 149-153: The prepared nanoparticles "show recognition performance comparable to that obtained via solid-phase synthesis." If it is comparable to the previous technique, what is the advantage of this method?
Line 154-155: "They also investigated the effect of the total monomers to template molar ratio on the formation of the binding sites, as well as, the pivotal role of hydrophobic interactions on 155 the formation of nanoscale particles." What did they find?
-Line 176: What is a virtual imprinting method? What is a pseudo-template?
-Line 202: "Moreover, when the target molecule is specifically recognized in the solution, the residual template molecule in the MIPs may fall off, resulting in experimental errors." I understand that some residual template might cause errors, but how does the presence of the target molecule cause the template to fall off?
Line 266: What is Acrydite?
-Figure 3: The order of a,b,c and d is illogical (they are mentioned in the text in a different order). c is never mentioned in the text.
-Line 298-303: This is hard to understand and could really benefit from a figure.
-Line 301: What is "a layer of fluorescence"? Also, is "the sugar molecule" referring to the disaccharide coated protein?
-Line 304-318: After reading this section, I still don't know what "prefixed and unfixed" means.
-Line 309-310: The figure reference seems to be in the wrong place.
-Line 313: What is the source of affinity between the particles and the protein template?
-Figure 5: Figure 5b is never mentioned in the text (I think it would be good to check the figure references through the entire manuscript).
Line 446-447: "In recent years, since the emergence of antibiotic resistance in clinical trials, it has been found that microorganisms pose a serious threat to human life and health." I don't see what this has to do with anything described in the manuscript.
-Line 534-535: "solid-phase imprinting can obtain high-affinity MIPs in different elution temperatures" Does this mean high affinity at all temperatures or some temperatures?
-Line 619: Azobenzene usually responds to UV light, not green light. Was a special azobenzene used here?
-Line 648: The hepatitis virus is not a molecule
Line 670-677: I don't understand this section, and a figure would really help here.
Author Response
Point 1: There are various spelling and grammar mistakes throughout the manuscript. Also the figures are often of rather poor image quality.
Author response 1: Thanks a lot for the reminding. We have carefully checked for spelling and grammar mistakes and corrected them. All the grammar or spelling corrections were marked up using the “Track Changes” function. The figures have all been replaced with images with high resolution.
Point 2: While the topic of the manuscript is on stimuli-responsive MIPs, and this subject is extensively discussed in section 2, the conclusions hardly mention stimuli-responsiveness. Instead, the conclusions section lists some general challenges in the field of MIPs. This could be improved.
Author response 2: Thanks for your suggestion. This article focuses on stimuli-responsive MIPs, so the second and third parts of this article were covered. Our statement in the conclusion was not clear enough. It should have listed the general challenges in the field of SR-MIP-NMs. We have modified it in the revised version.
Point 3: It is not always clear if the authors refer to polymers or polymer particles when they describe MIPs. When reading the beginning of section 1.2 I was initially confused if MIPs refers to dissolved polymers, polymer particles or bulk polymer materials. It does not help that the authors also use the word "bulk polymers" to describe the materials. This also leads to a weird situation in line 555 where it says "the ion concentration inside and outside the polymer will change", where I think "polymer particle" is intended.
Author response 3: As you might think, most of the MIPs for practice are polymeric particles. The bulk polymer materials described in Section 1.2 are MIPs prepared by traditional methods (e.g., bulk polymerization), while these polymers need to be grinded to small particles to be used. In this paper, we focus on molecularly imprinted nano- or micro-particles.
Point 4: Line 94: "Since MIP-NMs have comparable selectivity and affinity for natural receptors...". I doubt that "receptors" is the correct word here.
Author response 4: Thank you very much for your very valuable comments. The error has been corrected. We have changed it to "Since MIP-NMs have comparable selectivity and affinity to natural receptors...". The change was marked up using the “Track Changes” function.
Point 5: Line 126-128: Is the polymer on the magnetic nanoparticle a layer or particles? How is it bound to the nanoparticle?
Author response 5: Commonly, magnetic MIP nanoparticles were prepared by using the surface imprinting method, and by grafting a polymeric imprinting layer on the surface of magnetic nanoparticles. In this work, they synthesized superparamagnetic nanoparticles using a coprecipitation method and modified its surface with [3-(methacryloxy)propyl]trimethoxysilane (MPS). It forms end vinyl groups that provide sites for polymerization on the surfaces of nanoparticles. Finally, acrylamide and ethylene glycol dimethacrylate were copolymerized in the presence of (S)-naproxen on silica-coated iron oxide nanoparticles to mimic binding pockets resembling the size and shape of the template similar to an antibody-like biorecognition approach.
Some literatures using polymeric spheres mixed with magnetic nanoparticle, this is also a reliable method.
Point 6: Line 149-153: The prepared nanoparticles "show recognition performance comparable to that obtained via solid-phase synthesis." If it is comparable to the previous technique, what is the advantage of this method?
Author response 6: The advantage of this technique over previous techniques is that, like solid-phase imprinting method, each nanogel has on average only one imprinted site, which is similar to the interaction between receptor and ligand. It also has the advantage of better distribution, easier control, and specific recognition ability.
Point 7: Line 154-155: "They also investigated the effect of the total monomers to template molar ratio on the formation of the binding sites, as well as, the pivotal role of hydrophobic interactions on 155 the formation of nanoscale particles." What did they find?
Author response 7: Concerning the imprinting effect, the authors tested the recognition properties of nanoMIPs prepared with a wide range of TM: T molar ratios from 6.5:1 to 2400:1 (mol:mol). A parabolic profile was observed for the affinity of the imprinted binding sites respect to the molar ratio TM:T and the lowest Kd (1 nM) was reported for 175 ≤ TM: T ≤ 437. The influence of the monomer composition to the size of the resulting material was observed. Charge repulsion and hydrophobic effect contributed at most to the formation of the nanoMIPs (∼60 nm); and hydrophilic monomers yielded to sub-micron- to micron-sized particles.
Point 8: Line 176: What is a virtual imprinting method? What is a pseudo-template?
Author response 8: The virtual imprinting method is to select other substances with similar or the same specific structure as the target to replace the target as the template to prepare MIPs, which can largely solve the problems such as the template is not easy to obtain or expensive, and avoid the influence of template leakage on the results. Especially applicable to the target of high cost, infectious, flammable, explosive, easy to degrade and so on is not suitable for template molecules. A pseudo-template is a template molecule that uses another substance that is similar or identical in structure to the target. In this paper, the absence of a functional group in the glucose molecule that is able to form a strong ionic interaction with monomers containing an acidic or basic functionality makes the direct imprinting of glucose a tricky task; therefore, a dummy imprinting approach is favored. Glucuronic acid (GA) was selected as the functionalized dummy template to do a dummy imprinting. GA was used instead of glucose.
Point 9: Line 202: "Moreover, when the target molecule is specifically recognized in the solution, the residual template molecule in the MIPs may fall off, resulting in experimental errors." I understand that some residual template might cause errors, but how does the presence of the target molecule cause the template to fall off?
Author response 9: We are sorry for the unclear statement. Here we would like to indicate that the residual template molecules in the MIPs (especially large-size or bulk MIPs) may fall off during the specific recognition process. In fact, the presence of the target molecule cannot cause the template to fall off. This problem mostly occurs in large-size MIPs or bulk MIPs. Due to the slow diffusion, the residual template in the MIP gradually spread to the surface. This will lead to experimental errors under different conditions.
Point 10: Line 266: What is Acrydite?
Author response 10: Acrydite™ is an attachment chemistry based on an acrylic phosphoramidite that can be added to oligonucleotides as a 5’-modification. Its chemical formula was shown in Figure 3 (a).
Point 11: Figure 3: The order of a,b,c and d is illogical (they are mentioned in the text in a different order). c is never mentioned in the text.
Author response 11: Thank you very much for your very valuable comments. All mistakes have been corrected. All changes were marked up using the “Track Changes” function.
Point 12: Line 298-303: This is hard to understand and could really benefit from a figure.
Author response 12: The literature here is from 1999, which describes surface imprinted proteins. We cannot find a figure from the paper. This content is difficult for you to understand, perhaps because the expression is not clear enough, so we reorganized the language and modified it.
It was changed into "In 1999, Shi et al. proposed the plate surface imprinting method, which firstly adsorbed proteins on mica with high atomic content, then coated disaccharide molecules on proteins, and the two bonded by hydrogen bond. Then a thin layer of fluorescent polymer was polymerized on the surface of sugar molecules. Finally, mica was removed and protein molecules were dissolved. A polydisaccharide surface imprinted polymer with protein-shaped holes was generated".
Point 13: Line 301: What is "a layer of fluorescence"? Also, is "the sugar molecule" referring to the disaccharide coated protein?
Author response 13: Thank you very much for your question. "a layer of fluorescence" has been changed into " a thin layer of fluorescent polymer ". And "the sugar molecule" referring to the disaccharide coated protein. Thanks a lot.
Point 14: Line 304-318: After reading this section, I still don't know what "prefixed and unfixed" means.
Author response 14: "Prefixed and unfixed" means that the template molecules are fixed on the substrate before polymerization and the template molecules are added together during polymerization. These are two different surface imprinting methods.
Point 15: Line 309-310: The figure reference seems to be in the wrong place.
Author response 15: Thank you very much for your suggestions. The reference position of the diagram has been changed to the correct position.
Point 16: Line 313: What is the source of affinity between the particles and the protein template?
Author response 16: It is well acceptable that the complementary interactions between the amino group in the functional monomer and the carboxyl group in the template protein contribute greatly to the affinity. After the removal of the template molecule, there is a complementary imprinting cavity to the functional group of the template molecule, so the cavity can show affinity to the template molecule.
Point 17: Figure 5: Figure 5b is never mentioned in the text (I think it would be good to check the figure references through the entire manuscript).
Author response 17: Thank you very much for your suggestion. We have added some discussion into the revised version about Figure 5b. The specific content added was " Teixeira et al. designed the conformational epitopes of transforming growth factor-β3 (TGF-β3) and imprinted the surface onto polyacrylamide-based nanoparticles by re-verse microemulsion polymerization (Figure 5b) ". The added content was marked up using the “Track Changes” function.
Point 18: Line 446-447: "In recent years, since the emergence of antibiotic resistance in clinical trials, it has been found that microorganisms pose a serious threat to human life and health." I don't see what this has to do with anything described in the manuscript.
Author response 18: Thanks for your valuable advice, we have deleted this part.
Point 19: Line 534-535: "solid-phase imprinting can obtain high-affinity MIPs in different elution temperatures" Does this mean high affinity at all temperatures or some temperatures?
Author response 19: Thank you for question, "solid-phase imprinting can obtain high-affinity MIPs in different elution temperatures, "which means having a high affinity at a particular temperature. Our expression was not clear enough, which has been revised in the paper. Change into "For example, by adding NIPAAm to monomers used in the solid-phase imprinting method, MIPs with different affinities at different temperatures can be eluted by controlling the temperature, resulting in high affinity MIPs, which is due to the volume change of PNIPAAm at different temperatures".
Point 20: Line 619: Azobenzene usually responds to UV light, not green light. Was a special azobenzene used here?
Author response 20: In this paper, a hydrophilic green-light-responsive azobenzene derivative was used as the functional monomer.
Point 21: Line 648: The hepatitis virus is not a molecule
Author response 21: Thanks for your suggestion. We are sorry that the expression was not accurate enough. We have revised it.
Point 22: Line 670-677: I don't understand this section, and a figure would really help here.
Author response 22: Thank you very much for your suggestions. We've added figure here (Figure 8) to make it easier to understand.
Round 2
Reviewer 1 Report
Still, only about half of the descriptions in the Strimuli Responsive MIP are in the Strimuli Responsive MIP, and there is a discrepancy between the title and the content of the MIP. If you do not want to change the content, I recommend changing the title to match the content.
Author Response
Response to Reviewer 1 Comments
Point: Still, only about half of the descriptions in the Stimuli Responsive MIP are in the Stimuli Responsive MIP, and there is a discrepancy between the title and the content of the MIP. If you do not want to change the content, I recommend changing the title to match the content.
Author response: Thank you very much for your very valuable comments. We carefully considered the structure in the context of the full text and decided not to directly change the title, but we have modified the content of the article. For the overall content of the article, the first part was to introduce the background of MIP-NMs (MIP nanomaterials); the second part was to discuss different stimuli-responsive MIP-NMs; and in the last part, biomedical applications of smart MIP-NMs, were summarized, and most of this part focused on MIPs with stimuli responsiveness. In previous version, we did not highlight stimuli responsiveness, while in the new version we added some discussion to highlight the stimuli responsiveness and this will make the content more compatible with the title.
We categorized the responses induced by the synergistic interaction based on composite materials (magnetic, luminescent materials) with MIPs as another way of stimulus-response, and echoing the applications presented in Section 3. The specific content added is:
" All the above stimuli-responses are based on the themselves of MIPs, such as the swelling and shrinkage of functional skeletons, or deformation of imprinted sites under stimuli. Moreover, another kind of response is caused by synergies of other composite materials with MIPs. For example, magnetic Fe3O4 nanoparticle as the composite material is to endow magnetic response with MIPs. Luminous units can be used as the core of core-shell MIP-NPs, which can display different signals after templates are bound. These responses are common in biomedical applications, such as bioanalysis, bioimaging, tracking in vivo and so on. We will introduce them in detail in the following part."
We also added some discussion about stimuli responsiveness in the Section 3 to make the content of the article more relevant to the title. The specific content added is:
"SR-MIP-NMs are often used as biosensors in the applications of bioanalysis and diagnosis. As we know, biosensors are required to have intuitive signal changes, but traditional MIPs must be compounded with other materials that can provide signals to meet this requirement. Therefore, these kinds of composite MIPs will produce different signals before and after binding template molecules, which can be also regarded as stimuli-responsive MIPs."
"Similar to the above fields of biological applications, the luminescent properties of MIP-NMs come from fluorescence molecules, quantum dots, carbon dots, etc. But some of these luminous signals won’t be changed after templates are bound, of which are different from those of MIPs biosensors. This kind of SR-MIP-NMs provides a means of tracking MIPs."
"The responses of common SR-MIP-NMs are mainly reflected in the application of drug delivery. Because the skeletons of SR-MIP-NMs shrink or swell or biodegraded when their microenvironment changes, and drug can be held well or released fast."
" From the development of MIP-NMs in biomedicine, we find that MIP-NMs are closer to the function of biological antibodies, especially in disease intervention. Whether the material structure or physical and chemical properties, MIPs prepared via solid-phase synthesis are undoubtedly the best candidates. Those MIP-NMs with thermal responsiveness, we believe, are bound to have more breakthrough applications soon."
The changes were marked up using the “Track Changes” function.